# Future life expectancy in Europe taking into account the impact of smoking, obesity, and alcohol

**Fanny Janssen[1,2]\*, Anastasios Bardoutsos[2], Shady El Gewily[2], Joop De Beer[1]**

[1]Netherlands Interdisciplinary Demographic Institute - KNAW/University of Groningen, The Hague, Netherlands; [2]Population Research Centre, Faculty of Spatial Sciences, University of Groningen, Groningen, Netherlands

**Abstract** Introduction: In Europe, women can expect to live on average 82 years and men 75 years. Forecasting how life expectancy will develop in the future is essential for society. Most forecasts rely on a mechanical extrapolation of past mortality trends, which leads to unreliable outcomes because of temporal fluctuations in the past trends due to lifestyle 'epidemics'. Methods: We project life expectancy for 18 European countries by taking into account the impact of smoking, obesity, and alcohol on mortality, and the mortality experiences of forerunner populations. Results: We project that life expectancy in these 18 countries will increase from, on average, 83.4 years for women and 78.3 years for men in 2014 to 92.8 years for women and 90.5 years for men in 2065. Compared to others (Lee–Carter, Eurostat, United Nations), we project higher future life expectancy values and more realistic differences between countries and sexes. Conclusions: Our results imply longer individual lifespans, and more elderly in society. Funding: Netherlands Organisation for Scientific Research (NWO) (grant no. 452-13-001).

\*For correspondence:
janssen@nidi.nl

**Competing interests:** The authors declare that no competing interests exist.

## Introduction

Mortality projections are essential for forecasting how long people will live on average, for predicting the future extent of population ageing, for determining the sustainability of pension schemes and social security systems, for setting life insurance premiums, and for helping governments plan for the demand for services such as healthcare (*European Commission, 2009*). Recently, the societal and academic relevance of obtaining reliable and robust mortality forecasts has increased due to the revived debate on the existence and proximity of a limit to human life expectancy (*Fries, 1980*; *Manton et al., 1991*; *Couzin, 2005*; *de Beer et al., 2017*; *Olshansky and Carnes, 2019*; *Vaupel et al., 2021*); the linkage between life expectancy developments and retirement age in many European countries (*Carone et al., 2016*; *OECD, 2016*); the stagnation of increases in life expectancy in the United States, the United Kingdom, and other European countries since 2011 (*Murphy et al., 2019*; *Raleigh, 2019*); and the recent COVID-19 pandemic.

The growing relevance of obtaining reliable mortality forecasts has led to important advances in mortality forecasting by professionals and researchers from – among others – demography, actuarial sciences, and public health (*Janssen, 2018*). In the past, mortality forecasts were often based on expectations fuelled by the debate on the limit to life expectancy (*Oeppen and Vaupel, 2002*; *Booth and Tickle, 2008*). The majority of the currently existing mortality forecasting methods, including those used by statistical offices in Europe, are mainly based on extrapolations of past trends in age-specific mortality (*Booth and Tickle, 2008*; *Stoeldraijer et al., 2013a*). These extrapolative methods rely on the regularities observed in mortality trends over time and in the age patterns (*Booth and Tickle, 2008*). An extrapolative approach to mortality forecasting is considered more objective, easier to apply, and more likely to result in reliable forecasts than the previously employed

 

**eLife digest** On average, in Europe, men can currently expect to live till the age of 75 and women until they are 82. But what will their lifespans be in the next decades? Reliable answers to this question are essential to help governments plan for future health care and social security costs.

While medical improvements are likely to further extend lifespans, lifestyle factors can result in temporal distortions of this trend. Yet, most estimates of future life expectancy fail to consider changing lifestyles, as they only use past mortality trends in their calculations. This can make these projections unreliable: for example, increases in smoking rates among Northern and Western European men led to stagnating male life expectancies in the 1950s and 1960s, but these picked up again after smoking declined. The same pattern is showing for women, except it is lagging as they took up smoking later than men. Based simply on the extrapolation of past mortality trends, current projection models fail to consider the past and predicted modifications of life expectancy trends prompted by changing rates of health behaviours – such as increases followed by (anticipated) declines in alcohol consumption and obesity rates, similar to what was observed with smoking.

To produce a more reliable forecast, Janssen et al. incorporated trends in smoking, obesity, and alcohol use into life expectancy projections for 18 European countries. The predictions suggest that life expectancy for women in these countries will increase from 83.4 years in 2014 to 92.8 years in 2065. For men, it will also go up, from 78.3 to 90.5 years.

In the future, this integrative approach may help to track the effects of health-behaviour related prevention policies on life expectancy, and allow scientists to account for changes caused by the COVID-19 pandemic. In the meantime, these estimates are higher than those obtained using more traditional methods; they suggest that communities should start to adjust to the possibility of longer individual lifespans, and of larger numbers of elderly people in society.

expectation approaches and explanation approaches (mortality forecasting by cause of death or with an explanatory model) (*Booth and Tickle, 2008*). The stochastic Lee–Carter mortality projection methodology (*Lee and Carter, 1992*) has long been the benchmark extrapolative mortality forecasting method (see *Janssen, 2018*).

It is, however, increasingly being acknowledged that mechanically extrapolating past national mortality trends can result in unreliable outcomes (e.g., *Janssen and Kunst, 2007*; *Janssen et al., 2013*; *Bongaarts, 2014*; *Stoeldraijer, 2019*; *Li and Raftery, 2021*; *Blakely, 2018*). First, when past trends in mortality are non-linear, the projection outcome will depend to a large extent on the period on which the extrapolation is based (*Janssen and Kunst, 2007*; *Janssen et al., 2013*; *Stoeldraijer, 2019*). Second, extrapolative mortality forecasts made for individual populations often project unrealistic crossovers or large future differences in life expectancy between populations (*Li and Lee, 2005*; *Janssen et al., 2013*; *Hyndman et al., 2013*). For example, extrapolating the large mortality declines observed in Eastern European countries – which have historically had relatively low life expectancy, but have experienced improvements since 2005 (*Trias-Llimós et al., 2018*) – would result in long-term future life expectancy values being higher in Eastern than in Western European countries with historically high life expectancy values and more moderate mortality declines.

We argue that to obtain a reliable and robust mortality forecast, it is essential to distinguish between (1) the general and gradual long-term mortality decline due to socio-economic and medical progress that can be extrapolated into the future, thereby taking into account the mortality experiences of other countries, and (2) remaining factors that cause deviations from this general mortality decline as well as country and sex differences in this trend.

While life expectancy has increased overall as a result of socio-economic and medical progress (*Omran, 1998*; *Oeppen and Vaupel, 2002*; *Mackenbach, 2013*), past trends in mortality and life expectancy were characterised by periods of stagnation or even decline and exhibited large differences between countries and sexes (e.g., *Vallin and Meslé, 2004*; *Leon, 2011*). Lifestyle factors, particularly smoking, alcohol abuse, and obesity, contributed greatly to these deviations and differences (e.g., *Lhachimi et al., 2016*; *WHO, 2018a*; *Janssen et al., 2021*). For example, the decelerating life expectancy improvements among men in many north-western European countries in the 1950s and 1960s, and, in later decades, in other European countries and among women, can be

largely explained by the smoking epidemic (*Rostron and Wilmoth, 2011*; *Janssen et al., 2015*; *Lindahl-Jacobsen et al., 2016*). The pattern of a rapid increase and a subsequent decline in smoking prevalence occurred first among men in Anglo-Saxon countries, and was followed later in other countries and by women (*Lopez et al., 1994*). These developments resulted in similar patterns in smoking-attributable mortality about 30–40 years later (*Lopez et al., 1994*; *Janssen, 2020*). The large increases in alcohol consumption and alcohol-attributable mortality between 1990 and 2005 in Eastern Europe contributed to the stalling of life expectancy improvements in Eastern Europe in this period (*Trias-Llimós et al., 2018*). The more favourable alcohol-related trends in Eastern Europe since 2005 have contributed to the convergence in life expectancy levels between Eastern and Western Europe. The stagnation in life expectancy improvements since approximately 2011 in the United States and the United Kingdom, but also in other selected European countries (*Murphy et al., 2019*), can be partly attributed to the high obesity prevalence and obesity-attributable mortality in these countries (*Raleigh, 2019*; *Janssen et al., 2020a*; *Janssen et al., 2020b* ) and to the increasing impact on life expectancy levels of past large increases in obesity prevalence (*Vidra et al., 2019*).

Previous mortality forecasts that incorporated lifestyle information (*Bongaarts, 2006*; *Pampel, 2005*; *Wang and Preston, 2009*; *King and Soneji, 2011*; *Preston et al., 2014*; *French and O'Hare, 2013*; *Foreman et al., 2018*; *Janssen and Kunst, 2007*; *Janssen et al., 2013*; *Li and Raftery, 2021*) generally included information on only one lifestyle factor, usually smoking. Moreover, these forecasts employed very different techniques and ignored the mortality experiences of other countries. The previous so-called coherent or multi-population forecasting methods that take into account the mortality experiences of other countries (*Li and Lee, 2005*; *Hyndman et al., 2013*; *Enchev et al., 2017*; *Eurostat, 2020b*; *Bergeron-Boucher et al., 2018*) have, despite their growing popularity, so far been applied almost exclusively to all-cause mortality. The two previous projections that incorporated both elements simultaneously only included the effect of smoking (*Janssen et al., 2013*; *Li and Raftery, 2021*). The approach developed for this purpose by *Janssen et al., 2013* consists of the combination of a coherent projection of non-smoking-related mortality with an age-period-cohort projection of smoking-attributable mortality. This approach was applied to the Netherlands (*Janssen et al., 2013*) and was adopted by Statistics Netherlands as part of their official population forecast (*Stoeldraijer et al., 2013b*). *Li and Raftery, 2021* applied an amended version of this approach to 69 countries.

Our objective is to project future life expectancy in 18 European countries, while simultaneously taking into account the time-varying impact of smoking, obesity, and alcohol on mortality, and the mortality experiences of forerunner populations.

## Materials and methods

Our approach to mortality forecasting relied on an analysis of past mortality trends and their determinants. In line with the evidence gathered, we distinguished between (1) the general and gradual long-term mortality decline not affected by the three lifestyle factors that could be extrapolated into the future, while taking into account the mortality experiences of other countries, and (2) deviations from and differences in this general mortality decline caused predominantly by the time-varying impact of smoking, obesity, and alcohol on mortality, which required the use of more advanced projection techniques.

The projection involved four steps. First, we determined the long-term decline in mortality and life expectancy without the combined effect of smoking, obesity, and alcohol. To this end, we used existing recent age- and sex-specific estimates of mortality after excluding the combined impact smoking, obesity, and alcohol (*Janssen et al., 2021*), which we refer to as non-lifestyle-attributable mortality. Second, we projected this long-term underlying mortality decline into the future, while taking into account the mortality experiences of other populations. Specifically, we performed a Li–Lee coherent projection (*Li and Lee, 2005*) of non-lifestyle-attributable mortality in which we regarded the high life expectancy values and more favourable long-term trends among women in France, Spain, and Italy as the values and trends towards which the other populations will converge, since we regard these three populations as forerunners. Third, we obtained future estimates of mortality attributable to smoking, obesity, and alcohol, which we refer to as lifestyle-attributable mortality. We did so by utilizing recently published projections of smoking-, obesity-, and alcohol-attributable mortality that were both data and theory driven (*Janssen et al., 2020c*; *Janssen et al.,*

*2020b*; *Janssen et al., 2020d*). Fourth, we combined the projections of non-lifestyle-attributable mortality and lifestyle-attributable mortality by extending the approach that Janssen et al. developed for combining the separate projections of non-smoking- and smoking-attributable mortality (*Janssen et al., 2013*).

We applied our projection approach to the national populations of 18 European countries (see *Table 1* for the included countries) using country, sex, and age-specific all-cause mortality and exposure data for the 1990–2014 period from the Human Mortality Database (*HMD and University of California, Berkeley (USA), and Max Planck Institute for Demographic Research (Germany), 2019*), and similar data for lifestyle-attributable mortality and non-lifestyle-attributable mortality from a recent study (*Janssen et al., 2021*). We were restricted in both the starting year and the end year because lifestyle-attributable mortality data was available only for this period.

We obtained estimates of life expectancy at birth (e0), including 95% projection intervals up to 2065, by applying standard life table techniques to the projected mortality rates for ages 0–130 (*Preston et al., 2001*). We focussed on life expectancy at birth because it is a very common

**Table 1.** Comparison of observed gains in life expectancy at birth (e0) between 1990 and 2014 with the gains when the effects of smoking, obesity, and alcohol are removed ( = for non-lifestyle-attributable mortality), 18 European countries, by country and sex.

| | Gain e0 1990–2014 | | | |
| | Observed | | Non-lifestyle-attributable mortality | |
| Country | Men | Women | Men | Women |
|---|---|---|---|---|
| Austria | 6.69 | 4.87 | 5.05 | 5.29 |
| Belgium | 5.88 | 4.18 | 4.22 | 4.90 |
| Czech Republic | 8.19 | 6.33 | 5.50 | 6.62 |
| Denmark | 6.54 | 4.94 | 4.98 | 5.01 |
| Finland | 7.19 | 4.96 | 5.19 | 5.56 |
| France | 6.56 | 4.47 | 4.95 | 4.84 |
| Germany | 6.52 | 4.92 | 4.99 | 5.45 |
| Greece | 3.80 | 4.18 | 3.44 | 4.23 |
| Hungary | 7.12 | 5.47 | 5.61 | 6.16 |
| Ireland | 7.03 | 5.52 | 5.97 | 5.90 |
| Italy | 6.92 | 4.91 | 4.67 | 4.97 |
| Netherlands | 6.05 | 3.20 | 4.20 | 4.50 |
| Norway | 6.58 | 4.29 | 6.10 | 5.21 |
| Poland | 7.41 | 6.13 | 6.56 | 7.04 |
| Slovenia | 8.20 | 6.00 | 5.62 | 6.39 |
| Sweden | 5.54 | 3.66 | 5.09 | 4.35 |
| Switzerland | 6.98 | 4.40 | 5.25 | 4.76 |
| United Kingdom | 6.41 | 4.49 | 4.89 | 4.67 |
| *Average* | *6.64* | *4.83* | *5.13* | *5.33* |
| *Min* | *3.80* | *3.20* | *3.44* | *4.23* |
| *Max* | *8.20* | *6.33* | *6.56* | *7.04* |
| *Variance* | *0.92* | *0.66* | *0.51* | *0.62* |
| *Annual change* | *0.28* | *0.20* | *0.21* | *0.22* |
| *Forerunners** | *4.81* | *4.81* | *5.00* | *5.00* |

*French, Italian, and Spanish women (unweighted average). For Spanish women, the gain in e0 (1990–2014) was 5.05 years for all-cause mortality and 5.20 years for non-lifestyle-attributable mortality.

The online version of this article includes the following source data for Table 1:

**Source data 1.** Data behind *Table 1*.

summary measure of health, and the most common output measure of mortality forecasts. We have chosen a relatively large projection horizon (2015–2065) given the comparatively short historical time series available (1990–2014) to illustrate that our approach is able to generate reliable outcomes for the long-term future.

We compared our projection outcomes with the outcomes of the benchmark Lee–Carter extrapolation (*Lee and Carter, 1992*) applied to all-cause mortality and examined the separate effects of incorporating lifestyle factors, and of including the mortality experiences of other countries. We report the outcomes of additional projections and comparisons in the Results section 'The differences explained'. We also compared our outcomes to the official forecasts by Eurostat and United Nations (see the Discussion section 'Comparison with other projections').

More detailed information on the data and methods can be found in Appendix 1.

## Results

### Underlying long-term increase in life expectancy

Over the 1990–2014 period, life expectancy at birth (e0) increased, on average, across the 18 European countries studied by 6.6 years for men (from 71.8 to 78.3 years) and by 4.8 years for women (from 78.6 to 83.4 years) (see *Table 1*). These values translate into a yearly increase in e0 of 0.28 years for men and 0.20 years for women. This difference between men and women can largely be attributed to lifestyle factors. Without the impact of smoking, obesity, and alcohol, the increase in e0 over the 1990–2014 period was more similar between men (5.1 years; 0.21 annually) and women (5.3 years; 0.22 annually). In addition, the increase in e0 was more similar between countries for non-lifestyle-attributable mortality (variance: 0.5 for men, 0.6 for women) than for all-cause mortality (variance: 0.9 for men, 0.7 for women).

The greater increase in e0 for all-cause mortality than for non-lifestyle-attributable mortality among men can be explained by the significant declines in lifestyle-attributable mortality men experienced (see *Figure 1*). These declines stemmed predominantly from large declines in smoking-attributable mortality (*Figure 1—figure supplement 1*), after a period of sharp increases. In contrast, the past increases in lifestyle-attributable mortality among women were driven by increases in all three factors (except alcohol in Eastern Europe) (*Figure 1—figure supplement 2*) and resulted in smaller increases in e0 for all-cause mortality than for non-lifestyle-attributable mortality.

*Figure 2* and *Figure 2—figure supplement 1* show that whereas trends in e0 since 1950 have been rather unstable, the trends in e0 without lifestyle factors can be regarded as more stable. This is indicated by (1) the more stable trend in e0 for non-smoking-related mortality for men in the 1950s and 1960s, which seems to be in line with the more recent trends in e0 for non-lifestyle-attributable mortality and (2) the close correspondence among women between the recent trends in e0 for non-lifestyle-attributable mortality and the trends in e0 for all-cause mortality in the 1950s–1960s that were probably affected only minimally by lifestyle factors.

### Projected future level of life expectancy

We extrapolated the more universal and more stable mortality trends that we observed for non-lifestyle-attributable mortality, while assuming that the trends for men and women in the individual countries will eventually move towards the more favourable long-term trends for women in France, Spain, and Italy. We selected these populations as the forerunner populations in terms of life expectancy in Europe because they exhibit both very high recent e0 values and very favourable long-lasting past trends in e0 (*Stoeldraijer, 2019*).

We added to these projections the projected levels of lifestyle-attributable mortality (see *Figure 1*). For men, the past declines in age-standardised lifestyle-attributable mortality fractions (LAMF) over the 1990–2014 period are projected to further decline until 2065, albeit at a different pace than in the past. Among Eastern European men in particular, the decline is projected to decelerate. For women, we project that the past increases in LAMF will (eventually) turn into declines. *Figure 1—figure supplements 1* and *2* show the projections of smoking-, obesity-, and alcohol-attributable mortality (*Janssen et al., 2020c*; *Janssen et al., 2020b*; *Janssen et al., 2020d*). The projections follow the observed wave-shaped dynamic of the smoking epidemic (*Lopez et al., 1994*; *Janssen, 2020*) and the hypothesised wave shape of the obesity epidemic (*Xu and Lam,*

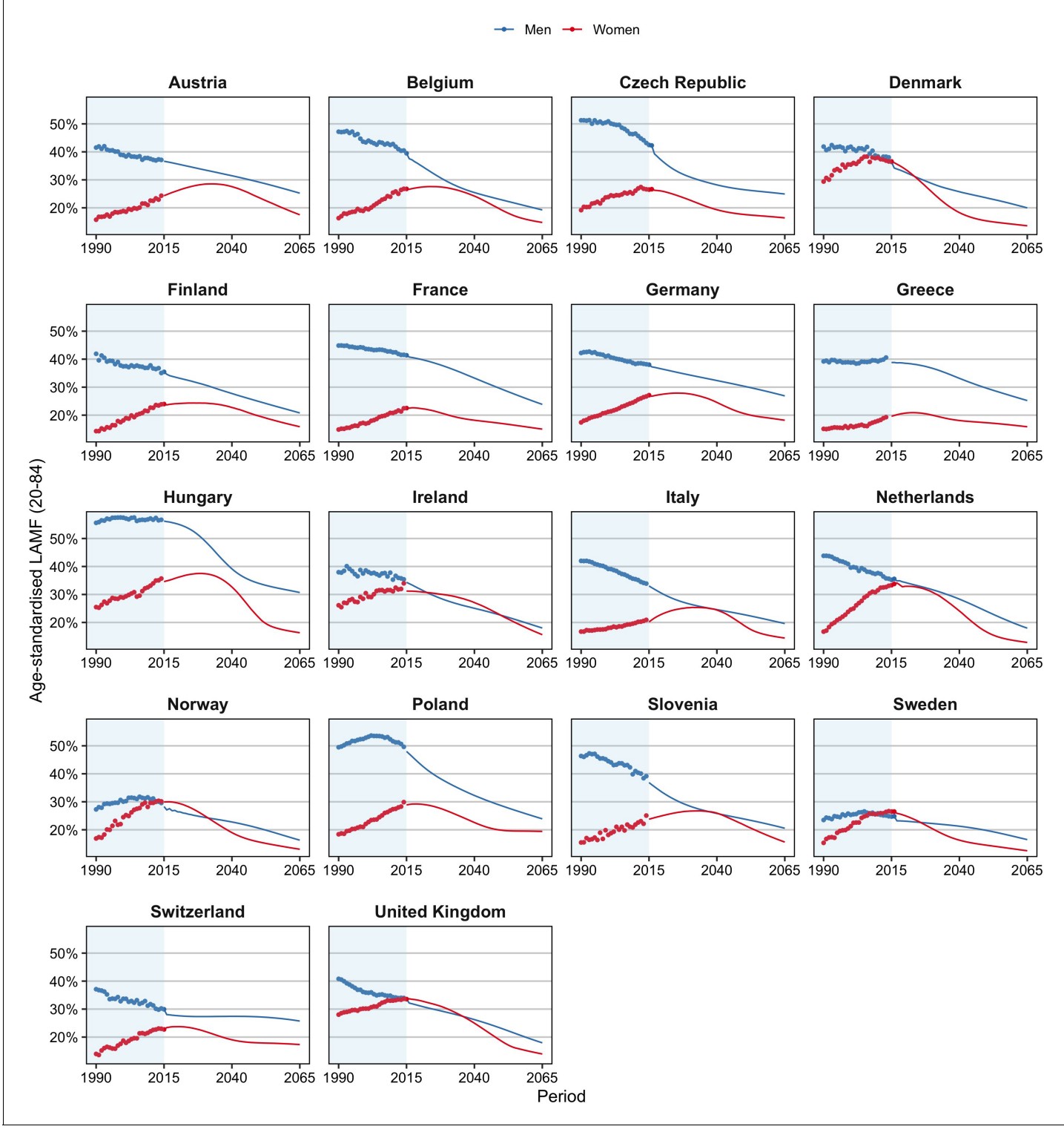

**Figure 1.** Past and projected age-standardised lifestyle-attributable mortality fractions (LAMF) (20–84), 1990–2065, by sex, for 18 European countries. The online version of this article includes the following source data and figure supplement(s) for figure 1:

**Source data 1.** Data behind *Figure 1*.

**Figure supplement 1.** Past and projected (=median) age-standardised lifestyle-attributable mortality fractions (20–84) (smoking, alcohol, obesity, combined), 1990–2065, by country, men.

**Figure supplement 1—source data 1.** Data behind *Figure 1—figure supplement 1*.

*Figure 1 continued on next page*

*Figure 1 continued*

**Figure supplement 2.** Past and projected (=median) age-standardised lifestyle-attributable mortality fractions (20–84) (smoking, alcohol, obesity, combined), 1990–2065, by country, women.

**Figure supplement 2—source data 1.** Data behind *Figure 1—figure supplement 2*.

*2018*; *Jaacks et al., 2019*). For alcohol-attributable mortality, unrealistic future differences between countries were avoided by assuming that the current increases observed for selected countries will eventually turn into declines (*Janssen et al., 2020d*).

We project that in the 18 European countries studied, life expectancy at birth will increase from, on average, 78.3 years for men and 83.4 years for women in 2014 to 90.5 years for men and 92.8 years for women in 2065 (*Table 2*; *Figure 3*). This represents an average increase per year of 0.24 years for men and 0.18 years for women. The projected increase is greater for men than for women because all-cause mortality levels are currently further away from non-lifestyle-attributable mortality levels for men than for women and because lifestyle-attributable mortality is higher among men than among women. There is, therefore, more room for improvement for men than for women.

Compared to the average past increases (1990–2014) in e0 (0.28 years for men; 0.20 years for women), these projected increases are lower, particularly for men. This is because – as mentioned – the increase in e0 for men over the 1990–2014 period is actually an acceleration compared to the underlying trend in non-lifestyle-attributable mortality because of the decline in lifestyle-attributable mortality, predominantly in smoking-attributable mortality.

The highest e0 in 2065 is projected for France among women (94.0) and for Italy among men (91.4). Hungary is the country with the lowest projected e0 in 2065, at 88.6 years among men and 91.0 years among women. These countries also had in 2014, respectively, the (second) highest and the lowest values. We project the largest gains in life expectancy for Hungary and Poland, particularly among men, in line with past large increases in non-lifestyle-attributable mortality due to a process of catch-up (*Table 1*), and large projected declines in lifestyle-attributable mortality in these countries (*Figure 1*). Among women, we project large gains as well for Denmark and the United Kingdom, which can be attributed to their high current levels of lifestyle-attributable mortality, predominantly due to their high levels of smoking-attributable mortality, and the large projected (continued) declines therein (*Figure 1—figure supplement 2*). The smallest gains are projected for Switzerland, which could be related to the relatively small projected declines in lifestyle-attributable mortality resulting from a long projected continuation of increases in obesity prevalence.

The sex difference in e0 is projected to decline from, on average, 5.0 years in 2014 to 2.3 years in 2065 (*Supplementary file 1A*; *Figure 3*). The gap is projected to be smallest in Greece, at 1.5 years; and largest in Slovenia, at 2.7 years. Among the explanations for the projected convergence of the sex gap in e0 is that the sex differences are smaller for non-lifestyle-attributable mortality than for all-cause mortality; the sex differences in lifestyle-attributable mortality are smaller in the long run; and our assumption that the trends in non-lifestyle-attributable mortality will converge.

## Comparison with the benchmark Lee–Carter mortality projection

Compared to the benchmark Lee–Carter (LC) extrapolative mortality projection, our projection results in higher future e0 in 2065 for all populations under study (*Table 2*). On average, the projected e0 values in 2065 are 2.6 years higher for men and 2.1 years higher for women in our projection than in the individual Lee–Carter mortality projection, which projected e0 values of 87.9 years for men and 90.6 years for women in 2065 (*Table 2*). For men, particularly in Hungary, the differences between our projected values and those of the LC projection are considerable, mainly because of large projected declines in lifestyle-attributable mortality. Also for Dutch women a considerable difference between our projection and the LC projection can be observed. For Dutch women not only large declines in lifestyle-attributable mortality are projected, but they also experienced a more favourable trend in non-lifestyle-attributable mortality than in all-cause mortality.

In addition, our projection results in smaller differences in future e0 between countries and sexes. That is, the LC projection projects that the variance between countries in e0 for 2065 is 3.8 years for men and 1.7 years for women, whereas in our projection, the expected variance is 0.6 years for both men and women (*Table 2*). In addition, *Figure 3—figure supplement 1* illustrates that the LC

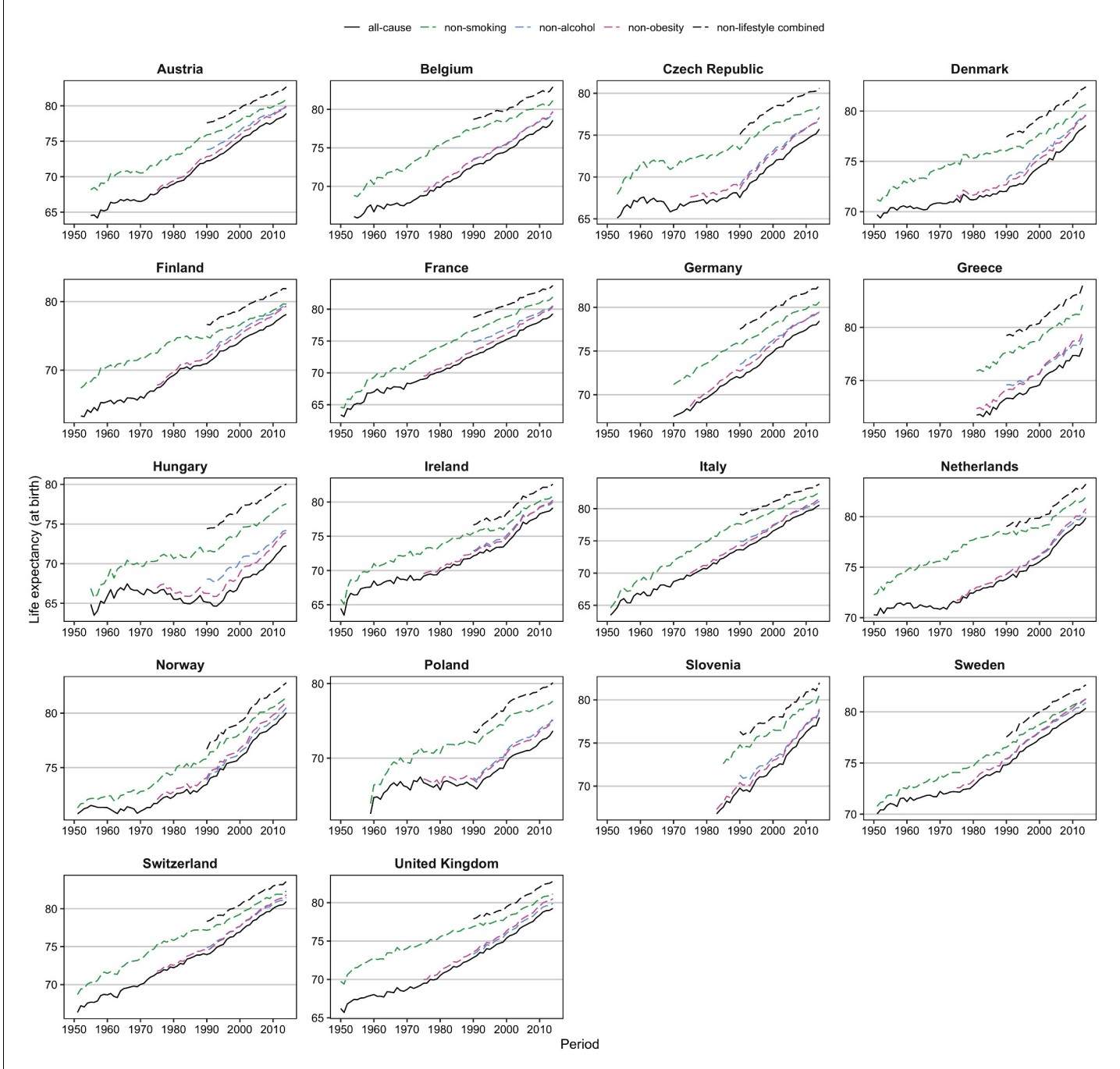

**Figure 2.** Comparison of trends in observed life expectancy at birth (e0) ( = all-cause) with trends in e0 when the effects of smoking, obesity and/or alcohol are removed ( = non-smoking, non-obesity, non-alcohol, non-lifestyle combined), 1950–2014*, men. *Based on the available information. Reproduced from various panels in Supplementary Figure 3a from *Janssen et al., 2021*, under the terms of a Creative Commons Attribution License (CC-BY 4.0; https://creativecommons.org/licenses/by/4.0/).

The online version of this article includes the following source data and figure supplement(s) for figure 2:

**Source data 1.** Data behind *Figure 2*.

**Figure supplement 1.** Comparison of trends in observed life expectancy at birth (e0) ( = all-cause) with trends in e0 when the effects of smoking, obesity and/or alcohol are removed ( = non-smoking, non-obesity, non-alcohol, non-lifestyle combined), 1950–2014*, women.

**Figure supplement 1—source data 1.** Data behind *Figure 2—figure supplement 1*.

**Table 2.** Observed (2014) and projected (2065) life expectancy at birth (e0), by country and sex, according to our projection methodology, which takes into account the impact of smoking, obesity, and alcohol and the mortality experiences of forerunner countries ('lifestyle and coherent'); the benchmark Lee–Carter extrapolative mortality projection applied to all-cause mortality ('Lee–Carter'); and when purely accounting for smoking, obesity, and alcohol ('adding lifestyle').

| | Men | | | | Women | | | |
| | | e0 2065 | | | | e0 2065 | | |
| Country | e0 2014 | Lee–Carter | Adding lifestyle | Lifestyle and coherent | e0 2014 | Lee–Carter | Adding lifestyle | Lifestyle and coherent |
|---|---|---|---|---|---|---|---|---|
| Austria | 78.91 | 88.91 | 89.02 | 90.38 | 83.73 | 91.42 | 92.57 | 92.54 |
| Belgium | 78.57 | 87.76 | 88.20 | 90.93 | 83.52 | 90.09 | 92.43 | 92.98 |
| Czechia | 75.72 | 87.77 | 87.23 | 88.86 | 81.73 | 90.87 | 92.28 | 91.37 |
| Denmark | 78.56 | 87.72 | 88.48 | 90.50 | 82.67 | 89.66 | 93.33 | 93.08 |
| Finland | 78.13 | 88.25 | 88.30 | 90.22 | 83.85 | 90.90 | 92.65 | 92.69 |
| France | 79.28 | 89.11 | 90.03 | 91.40 | 85.44 | 92.77 | 94.24 | 94.01 |
| Germany | 78.43 | 88.36 | 88.67 | 90.18 | 83.35 | 90.78 | 92.60 | 92.62 |
| Greece | 78.46 | 85.07 | 86.32 | 90.97 | 83.83 | 89.75 | 90.51 | 92.44 |
| Hungary | 72.26 | 81.74 | 85.89 | 88.64 | 79.24 | 87.09 | 90.21 | 90.96 |
| Ireland | 79.15 | 89.59 | 90.57 | 90.95 | 83.23 | 91.67 | 94.38 | 93.40 |
| Italy | 80.55 | 89.25 | 89.14 | 91.43 | 85.16 | 92.30 | 93.34 | 93.53 |
| Netherlands | 79.87 | 88.67 | 88.73 | 91.11 | 83.29 | 88.80 | 92.59 | 93.18 |
| Norway | 80.03 | 88.86 | 90.23 | 90.87 | 84.09 | 90.72 | 93.48 | 93.35 |
| Poland | 73.66 | 85.21 | 88.28 | 89.37 | 81.41 | 90.44 | 92.74 | 91.88 |
| Slovenia | 77.96 | 88.68 | 88.95 | 89.91 | 83.69 | 92.20 | 93.79 | 92.60 |
| Sweden | 80.35 | 88.37 | 89.18 | 90.64 | 84.05 | 89.92 | 92.27 | 93.01 |
| Switzerland | 80.93 | 89.77 | 89.31 | 90.88 | 85.11 | 91.41 | 92.70 | 93.16 |
| United Kingdom | 79.25 | 88.99 | 89.35 | 91.04 | 82.99 | 89.93 | 92.70 | 93.34 |
| *Average* | *78.34* | *87.89* | *88.66* | *90.46* | *83.35* | *90.59* | *92.71* | *92.78* |
| *Min* | *72.26* | *81.74* | *85.89* | *88.64* | *79.24* | *87.09* | *90.21* | *90.96* |
| *Max* | *80.93* | *89.77* | *90.57* | *91.43* | *85.44* | *92.77* | *94.38* | *94.01* |
| *Variance* | *4.98* | *3.77* | *1.41* | *0.62* | *2.02* | *1.73* | *1.07* | *0.55* |

The online version of this article includes the following source data for Table 2:

Source data 1. Data behind *Table 2*.

projection, unlike our projection, can result in unlikely crossovers and divergences between countries (e.g., Greece and Hungary for men; the Netherlands and Slovenia for women).

In addition, the sex difference in e0 in 2065 is projected to be 2.7 years in the LC projection, but 2.3 years in our projection (*Table 2*). Furthermore, a closer look at the sex differences projected by the LC projection (*Supplementary file 1A*) reveals large differences in the projected sex differences between countries ranging from 0.1 to 5.4 years. In our projection, by contrast, the range is between 1.5 and 2.7 years. The very small difference in e0 between women and men projected for 2065 in the Netherlands (0.1 years) and in the United Kingdom (0.9 years) indicates that the individual LC projection of all-cause mortality might result in an unlikely crossover between men and women in e0 in the long-term future due to the more favourable past trends in all-cause mortality among men than among women in the 1990–2014 period. Thus, our projection avoids an unlikely crossover in future life expectancy values between men and women.

Moreover, unlike the LC mortality projection, our projection results in non-linear future increases in e0 (*Figure 3*; *Figure 3—figure supplement 2*). Particularly for men in Belgium, Greece, the Czech Republic, Hungary, and Poland, but also for women in Hungary, Denmark, and the United Kingdom, we see faster projected increases in life expectancy in the early decades, but slower increases in life

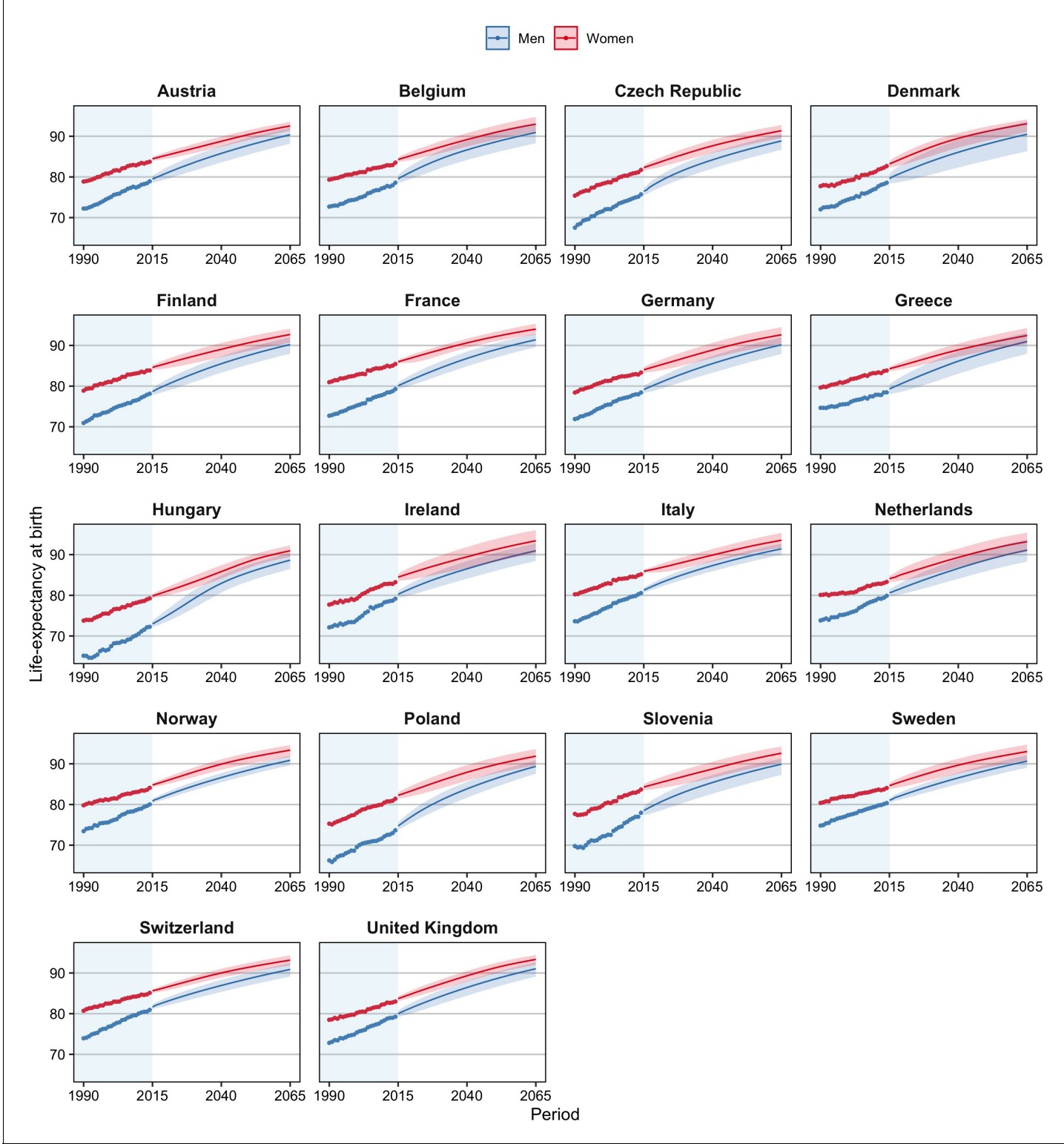

**Figure 3.** Observed and projected life expectancy at birth (including 95% projection intervals) using our projection approach, which takes into account the impact of smoking, obesity, and alcohol on past and future mortality trends, and the mortality experiences of forerunner countries, 18 European countries, by country and sex, 1990–2065.

The online version of this article includes the following source data and figure supplement(s) for figure 3:

**Source data 1.** Data behind *Figure 3*.

*Figure 3 continued on next page*

*Figure 3 continued*

**Figure supplement 1.** Comparison between countries of observed and projected life expectancy at birth, for our projection approach and the Lee–Carter extrapolation of all-cause mortality, 18 European countries, by sex, 1990–2065.

**Figure supplement 1—source data 1.** Data behind *Figure 3—figure supplement 1*.

**Figure supplement 2.** Observed and projected life expectancy at birth (e0), by country and sex, 1990–2065, for our projection approach, which takes into account the impact of smoking, obesity, and alcohol and the mortality experiences of forerunner countries, and for the Lee–Carter mortality projection applied to all-cause mortality.

**Figure supplement 2—source data 1.** Data behind *Figure 3—figure supplement 2*.

**Figure supplement 3.** Observed and individually projected life expectancy at birth (e0) for all-cause mortality, non-lifestyle-attributable mortality, and the combined projection of lifestyle- and non-lifestyle-attributable mortality, 18 European countries, by sex, 1990–2065.

**Figure supplement 3—source data 1.** Data behind *Figure 3—figure supplement 3*.

expectancy further in the future, in line with the projections of lifestyle-attributable mortality. The LC projection, by contrast, anticipates more constant future increases in life expectancy at birth.

## The differences explained

The observed differences between our projection and the Lee–Carter projection can stem from either the lifestyle dimension, the coherent dimension, or both. Within the lifestyle dimension, two mechanisms apply: first, the projection of non-lifestyle mortality instead of all-cause mortality, and, second, the effect of projected lifestyle-attributable mortality itself.

The higher values estimated by our projection than by the LC projection stem from both dimensions (*Table 2*; *Supplementary file 1B*). The coherent dimension led to higher values because the French, Italian, and Spanish women combined exhibited more favourable trends in non-lifestyle-attributable mortality than the individual populations combined, after controlling for the underlying age pattern of mortality. The lifestyle dimension resulted in an additional effect, which was particularly large for women. For women, unlike for men, the past trends in non-lifestyle-attributable mortality were more favourable than the past trends in all-cause mortality. For both men and women, an additional positive effect emerges from the projected (eventual) decline in lifestyle-attributable mortality.

The smaller projected differences between countries in our projection than in the LC projection also stem from both elements, although for men, the impact of integrating lifestyle is more pronounced (*Table 2*; *Supplementary file 1B*). Particularly for men, the past trends in non-lifestyle-attributable mortality were much more similar between countries than the past trends in all-cause mortality. In addition, for both men and women, country differences in lifestyle-attributable mortality in 2065 are projected to be smaller than they are currently.

The smaller sex differences in projected e0 values in our projection stem purely from the convergence dimension, whereas the nonlinearity of future trends in our projection stems completely from the lifestyle dimension. *Figure 3—figure supplement 3* illustrates that the inclusion of the lifestyle dimension in the Lee–Carter projection (thus ignoring the coherent dimension) leads to a future e0 that is moving back upwards towards the future trend in e0 for non-lifestyle-attributable mortality.

## Discussion

### Appraisal of our projection methodology

An important element of our projection approach is the identification of the underlying long-term mortality trend that can be extrapolated into the future. We illustrated that past trends in non-lifestyle-attributable mortality serve this purpose better than past trends in all-cause mortality. For non-lifestyle-attributable mortality, the differences in past trends between countries and between men and women are smaller, and the trends are more robust. In contrast, the trends in all-cause mortality exhibit a wave-shaped fluctuation due to initial increases in smoking-, obesity-, and alcohol-attributable mortality, followed (eventually) by declines. This pattern leads, first, to a deceleration of increases in e0, and a move away from the underlying increase in e0 in non-lifestyle attributable mortality, followed by an acceleration of increases in e0, and a move back towards the underlying increase in e0 in non-lifestyle-attributable mortality. Thus, extrapolating past declines in all-cause

mortality will lead to non-reliable outcomes: i.e., the outcomes will be either too low when decelerating increases are extrapolated or too high when accelerating increases are extrapolated. Similarly, the extrapolation of past trends in all-cause mortality will result in non-robust outcomes, when updating the projection after a few years. Because the past trends in non-lifestyle-attributable mortality are more linear than the past trends in all-cause mortality, especially from a historical point of view, our projection is less dependent on the data period used for the extrapolation.

Our finding of an average annual past increase in e0 for non-lifestyle-attributable mortality of 0.21 years for men and 0.22 years for women across the 18 European countries over the 1990–2014 period is lower than the continuous upward trend in record life expectancy of 0.243 years that *Oeppen and Vaupel, 2002* observed over the 1840–2000 period. There are two main reasons. First, whereas record values always increase, the levels in individual countries may not, and since the 1990s, the record life expectancy has been observed for a non-European country (Japan). Second, more recent increases in life expectancy are driven more by declines in mortality from chronic diseases at older ages with a less strong effect on life expectancy than from declines in mortality from infectious diseases at younger ages, which drove increases in life expectancy up to 1950 (*Omran, 1998*; *Vallin and Meslé, 2009*). As a result, a deceleration of increases in life expectancy over time is inevitable. Nonetheless, these large past increases in e0 for non-lifestyle-attributable mortality indicate that considerable increases in e0 are likely in the future.

We applied a coherent mortality projection approach to past trends in non-lifestyle-attributable mortality. This resulted in more realistic estimates of the differences in future e0 values between countries and sexes than those of the individual Lee–Carter projection of past trends. Another added value of our inclusion of the mortality experiences of other countries in our projection is that it provided a broader empirical basis for the determination of the most likely future developments (*Janssen and Kunst, 2007*), which, in turn, improved the robustness of our projection outcomes (*Stoeldraijer, 2019*). However, next to a potential effect on the outcomes of the exact coherent methodology that is applied (*Stoeldraijer, 2019*), it is important to take into account the choice of the reference population (*Stoeldraijer, 2019*; *Booth and Tickle, 2008*). Using forerunner populations as the reference population, as advocated by among others *Oeppen and Vaupel, 2002* and *Bergeron-Boucher et al., 2018*, instead of an average across many populations (*Li and Lee, 2005*; *Eurostat, 2020a*) will generally result in higher future life expectancy values (see the next section as well). However, given that the past trend in e0 for non-lifestyle-attributable mortality is more equal between countries and sexes than the past trend in e0 for all-cause mortality (see *Table 1*), which reference populations are chosen will likely have a larger impact when coherently forecasting all-cause mortality than when coherently forecasting non-lifestyle-attributable mortality (see also *Janssen and Kunst, 2007* and *Janssen et al., 2013*). In addition, our convergence assumption relies on – at least – a continuation of improvements in socio-economic developments and medical care. Such continued improvements depend on continued investments in these areas, including diminishing socio-economic inequalities, better preparations for additional possible outbreaks of infectious diseases, and for the potential harmful effects of climate change. Although this might be considered optimistic, our assumption is supported by evidence that life expectancy has long followed a continuously increasing trend (*Oeppen and Vaupel, 2002*), and that this trend is even stronger from a historical perspective if the effects of lifestyle factors are excluded (see the description of Figure 2 and its supplement in the Results section).

To the projected long-term trend in non-lifestyle-attributable mortality, we added estimates of how smoking-, obesity-, and alcohol-attributable mortality are likely to develop in the near future – and, consequently, how long it will take before all-cause mortality trends again follow more closely the underlying trends in non-lifestyle-attributable mortality. We generated these estimates based on previously projected age-specific smoking-, obesity-, and alcohol-attributable mortality (*Janssen et al., 2020c*; *Janssen et al., 2020b*; *Janssen et al., 2020d*). To avoid generating unrealistic outcomes for the (long-term) future (e.g., continuing increases in smoking-attributable mortality for women, in obesity-attributable mortality, and in alcohol-attributable mortality in the few Eastern European countries where it is still increasing), the projections were based on advanced approaches that did not simply extrapolate past trends, but also took into account knowledge about the future progression of smoking-, alcohol-, and obesity-attributable mortality. Nonetheless, the projection of lifestyle-attributable mortality involves more subjectivity than the extrapolation of non-lifestyle-attributable mortality (see *Janssen et al., 2020c*; *Janssen et al., 2020b*; *Janssen et al., 2020d* for a

critical evaluation). For example, in the separate projections of smoking-, obesity-, and alcohol-attributable mortality, population-specific lower bounds were implemented to avoid unlikely future estimates of zero smoking, obesity, and alcohol prevalence, in line with the theory (*Lopez et al., 1994*; *Xu and Lam, 2018*), and to prevent unlikely crossovers between sexes and between countries in alcohol-attributable mortality. This approach could be considered conservative. On the other hand, however, the use of a wave-shaped pattern for obesity, and the large declines in alcohol-attributable mortality we project for Eastern European countries, could be considered too optimistic.

However, in our view, the added value of integrating lifestyle into mortality projections outweighs the uncertainties that come with it. First, trends in non-lifestyle-attributable mortality provide a better approximation of the underlying mortality trend. Second, our projection is driven not only by data, but also by theory, and is preceded by a careful study of past trends. Third, by distinguishing between the underlying mortality trend and the additional effect of the three lifestyle factors combined, these projections provide much greater transparency and insight than a mere mechanical extrapolation of past trends could. Fourth, our approach resulted in more realistic differences between countries and sexes than those projected by other approaches (see the next section as well). The abovementioned four added values of our mortality projection approach can be clearly linked to important criteria for evaluating the performance of mortality forecasts in addition to accuracy: namely, robustness, plausibility, reasonableness, and transparency (*Cairns et al., 2009*). Moreover, the outcomes of our projection approach – which rely above all on the coherent projection of non-lifestyle-attributable mortality – are less dependent on the explicit choices that underlie general mortality forecasts (e.g., the choice of the forerunner populations and the historical time period).

## Comparison with other projections

Compared to the benchmark Lee–Carter mortality extrapolation, our projection approach results in higher e0 in the long run; smaller differences in future e0 between countries and sexes; and non-linear future increases in e0, in line with the shift towards the higher e0 values for non-lifestyle-attributable mortality.

Moreover, compared to the most recent official mortality forecast by *Eurostat, 2020a* and the *United Nations, 2020*, our mortality projection resulted in substantially higher projected e0 values in 2065 (*Supplementary file 1C*). Our values were, on average, 5.0 years higher for men and 3.2 years (Eurostat) and 4.3 years (UN) higher for women. The UN projects very large and unrealistic differences in future life expectancy between countries and sexes, including higher projected life expectancy for men than for women in a number of countries. These outcomes clearly show the added value of adopting a coherent mortality projection approach. While Eurostat has performed a coherent mortality projection, it used a different methodology. Eurostat made a partial convergence assumption, applied to men and women separately, and considered the aggregate all-cause mortality of 12 countries (Belgium, Denmark, Germany, Spain, France, Italy, the Netherlands, Austria, Portugal, Finland, Sweden, and the United Kingdom) as the target (*Lanzieri, 2009*; *Eurostat, 2020b*). In addition, Eurostat did not integrate the role of lifestyle factors.

The added value of our projection approach over pure coherent mortality forecasting is that we projected the more robust past trends in non-lifestyle-attributable mortality rather than in all-cause mortality, and we projected non-linear future trends in line with the projection of lifestyle-attributable mortality. The reason why coherent mortality projections have to be applied to non-lifestyle-attributable mortality rather than to all-cause mortality is that there are important differences in smoking-, obesity-, and alcohol-attributable mortality trends between countries and sexes (*Janssen, 2020*; *Vidra et al., 2019*; *Janssen et al., 2020d*).

Among the advantages of taking the effects of obesity and alcohol in addition to those of smoking into account is that e0 values for non-lifestyle-attributable mortality are higher than e0 values for non-smoking-attributable mortality (*Figure 2*, *Figure 2—figure supplement 1*), and thus have the potential for larger future increases. In addition, as was illustrated, the effects of both obesity-attributable mortality and alcohol-attributable mortality (particularly among Eastern European men) on past and future life expectancy trends are large.

## Overall conclusion and implications

All in all, the outcomes of our projection approach, which simultaneously takes into account the lifestyle factors smoking, obesity and alcohol, and the mortality experiences of forerunner populations, can be considered more realistic, more robust, and more insightful than previous extrapolative mortality projections, including those by the United Nations and Eurostat.

Our approach, which distinguishes between the underlying long-term mortality decline and the remaining factors that cause deviations from this mortality decline, could also be adopted to account for the impact of the COVID-19 pandemic on mortality and life expectancy. First, it would be illogical to extrapolate the mortality trends up to 2020 to obtain life expectancy estimates for 2021 and beyond, because life expectancy in 2020 has been importantly negatively affected by COVID-19 (*Marois et al., 2020*), and will likely to continue to do so for at least 2021. This effect cannot be predicted using a simple extrapolation approach. Second, the underlying long-term mortality decline is argued not to be affected by COVID-19 (e.g., *Stoeldraijer, 2020*), in line with the predominant temporal dip in life expectancy as a result of the Spanish flu in 1918 (see *Roser, 2020*). Although, as mentioned, the continuation of the underlying long-term mortality decline does depend on important investments to achieve continued improvements in socio-economic developments and medical care.

Our findings of higher projected life expectancy values than those estimated by the benchmark Lee–Carter mortality projection and the official mortality forecasts by Eurostat and the United Nations have important implications for society. First, individuals could consider planning their life course (e.g., education, work) differently given their longer expected lifespan. Second, a higher projected life expectancy suggests that there will be more older people in the population, which will likely bring both opportunities (e.g., an experienced workforce) and challenges in terms of healthcare planning and social security. Our results imply that either the pension age will have to be raised or, if it is not, that the capital requirements of pension providers will need to be increased. For determining the pension age based on projected life expectancy, a projection that is more robust will have significant added value. More generally, having a more stable projection of mortality will allow for more robust population forecasts.

In addition, our findings of (1) higher projected life expectancy values than those currently used in most mortality projection methodologies and (2) of a rather robust long-term increase in e0 after the effects of smoking, obesity, and alcohol are controlled for, indicate that a limit to human life expectancy (average lifespan) is not within reach – an issue that continues to spark scientific debate (see the first paragraph of the introduction).

## Data sharing

Some of our original data regarding lifestyle-attributable mortality were based on previous publications, which, in turn, used data that are openly available. The all-cause mortality data and the exposure data can be obtained through the Human Mortality Database. We have provided source data files for all our tables and figures. These comprise the numerical data that are represented in the different figures, and the output on which the different tables are based. In addition, one excel file with all the final numerical/output data that were used as input for the tables and figures will be made available at the Open Science Framework: https://osf.io/ghu45/. In addition, we will upload there the underlying observed age-specific mortality rates (all-cause mortality, non-lifestyle-attributable mortality, lifestyle-attributable mortality) as well as the adjusted and projected age-specific mortality rates (medians and 90% and 95% projection intervals). The different R codes used for the different steps of the data analyses will be shared – as well – through the Open Science Framework link above.

## Acknowledgements

We thank Mark van der Broek (Research Master Economics, University of Groningen) for creating the final figures in R.

## Additional information

### Funding

| Funder | Grant reference number | Author |
|---|---|---|
| Netherlands Organisation for Scientific Research | Innovational Research Incentives Scheme Vidi/452-13-001 | Fanny Janssen Anastasios Bardoutsos Shady El Gewily |
| Netherlands Organisation for Scientific Research | Innovational Research Incentives Scheme Vici/VI.C.191.019 | Fanny Janssen |

The funders had no role in study design, data collection and interpretation, or the decision to submit the work for publication.

### Author contributions

Fanny Janssen, Conceptualization, Supervision, Funding acquisition, Validation, Investigation, Visualization, Methodology, Writing - original draft, Project administration, Writing - review and editing; Anastasios Bardoutsos, Formal analysis, Validation, Investigation, Methodology, Writing - review and editing; Shady El Gewily, Formal analysis, Writing - review and editing; Joop De Beer, Validation, Investigation, Writing - review and editing

### Author ORCIDs

Fanny Janssen https://orcid.org/0000-0002-3110-238X

### Decision letter and Author response

Decision letter https://doi.org/10.7554/eLife.66590.sa1
Author response https://doi.org/10.7554/eLife.66590.sa2

## Additional files

### Supplementary files

• Supplementary file 1. (A) Sex differences in observed (2014) and projected (2065) life expectancy at birth (e0), by country and sex, according to our projection methodology, which takes into account the impact of smoking, obesity, and alcohol and the mortality experiences of forerunner countries ('lifestyle and coherent'); the benchmark Lee–Carter extrapolative mortality projection applied to all-cause mortality ('Lee–Carter'); and when purely accounting for smoking, obesity, and alcohol ('adding lifestyle'). (B) Observed (2014) and projected (2065) life expectancy at birth (e0) according to different methodologies, by country and sex. (C) Comparison of projected life expectancy in 2065 between our final projection ('we') and the most recent projections by Eurostat and the United Nations (UN), by country and sex.

• Transparent reporting form

### Data availability

Some of our original data regarding lifestyle-attributable mortality were based on previous publications, which, in turn, used data that are openly available. The all-cause mortality data and the exposure data can be obtained through the Human Mortality Database. We have provided source data files for all our tables and figures. These comprise the numerical data that are represented in the different figures, and the output on which the different tables are based. In addition, at the Open Science Framework (https://osf.io/ghu45/) we uploaded (i) one excel file with all the final numerical / output data that were used as input for the tables and figures, (ii) the underlying observed age-specific mortality rates (all-cause mortality, non-lifestyle-attributable mortality, lifestyle-attributable mortality) as well as the adjusted and projected age-specific mortality rates (medians and 90% and 95% projection intervals), and (iii) the different R codes used for the different steps of the projection.

The following dataset was generated:

| Author(s) | Year | Dataset title | Dataset URL | Database and Identifier |
|---|---|---|---|---|
| Janssen F | 2020 | Future life expectancy in Europe taking into account the impact of smoking, obesity and alcohol | https://osf.io/ghu45/ | Open Science Framework, ghu45 |

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

# Appendix 1

## Supplementary data and methods

### Setting

We included data for the national populations, by sex and age, for 18 European countries, over the period 1990 up to 2014: Denmark, Finland, Norway, Sweden, Austria, Belgium, France, Germany, Ireland, the Netherlands, Switzerland, the United Kingdom, Greece, Italy, the Czech Republic, Hungary, Poland, and Slovenia.

We were restricted to the period 1990–2014 because data on alcohol-attributable mortality were only available from 1990 onwards, which meant, in turn, that we only had data on lifestyle-attributable mortality from 1990 onwards (see *Janssen et al., 2021*). We included the abovementioned European countries because realistic projections for smoking-, obesity-, and alcohol-attributable mortality were generated for these countries (see *Janssen et al., 2020c*; *Janssen et al., 2020b*; *Janssen et al., 2020d*).

### Data

For the abovementioned 18 European countries, for the years 1990 up to 2014, we obtained all-cause mortality (both death numbers and rates) and exposure data by country, sex and single year of age (0–100) from the *HMD and University of California, Berkeley (USA), and Max Planck Institute for Demographic Research (Germany), 2019* (update May 1 2019). In addition, we used estimated lifestyle-attributable mortality fractions by country, sex, and single year of age (0–100) from a recent study (*Janssen et al., 2021*). These lifestyle-attributable mortality fractions represent the share of mortality that can be attributed to smoking, obesity, and alcohol. Multiplying these fractions to the age- and sex-specific all-cause mortality rates, we obtained the age- and sex-specific lifestyle-attributable mortality rates. Subtracting these lifestyle-attributable mortality rates from all-cause mortality rates resulted in non-lifestyle-attributable mortality rates, which – consequently – refers to mortality that is not attributable to smoking, obesity, and alcohol.

In addition, we used for our analysis already published projections of age-specific smoking-, obesity-, and alcohol-attributable mortality fractions for ages 0–84 (*Janssen et al., 2020c*; *Janssen et al., 2020b*; *Janssen et al., 2020d*).

### Overall projection approach

Our approach to mortality forecasting relied on an analysis of past mortality trends and their determinants. In line with the evidence gathered, we distinguished between (1) the general and gradual long-term mortality decline not affected by the three lifestyle factors that could be extrapolated into the future, while taking into account the mortality experiences of other countries, and (2) deviations from and differences in this general mortality decline caused predominantly by the time-varying impact of smoking, obesity, and alcohol on mortality, which required the use of more advanced projection techniques.

More specifically, our mortality approach consists of the combination of two projections:

1. A projection of non-lifestyle-attributable mortality using the coherent Li–Lee (LL) projection methodology, which takes into account the mortality experiences of other populations; and.
2. A projection of lifestyle-attributable mortality fractions based on already published projections of smoking-, obesity-, and alcohol-attributable mortality fractions.

### Main outcome measure

Our main outcome measure is life expectancy at birth (e0) by country and sex, which we estimated for the period 1990 up until 2014, and which we projected up to 2065. Life expectancy at birth is a very important summary measure of health. In addition, it is the most common summary output of mortality forecasts, although the reporting of other summary output measures, such as lifespan disparity, has recently attracted attention (*Bohk-Ewald et al., 2017*). We have chosen a relatively long

projection horizon (2015–2065) given the comparatively short historical time series available (1990–2014) to illustrate that our approach is able to generate reliable outcomes for the long-term future.

To obtain life expectancy at birth, we applied standard life table techniques (*Preston et al., 2001*) to either the observed or the projected age-specific mortality rates (0–130) for the different national populations under study. In doing so, we used the exponential method to transfer the age-specific mortality rates (mx) into age-specific probabilities (qx), by using the formula qx=1-exp(-mx).

To obtain age- and sex-specific all-cause and non-lifestyle-attributable mortality rates for ages 0–130 to be used as input in our life table calculations, we obtained the respective mortality rates for ages 101–130 by applying the Kannisto model of old-age mortality (*Thatcher et al., 1998*) to mortality for ages 80–100 for all-cause mortality and non-lifestyle-attributable mortality, respectively.

We compared our projection outcomes foremost with the outcomes of a Lee–Carter extrapolation (*Lee and Carter, 1992*) applied to all-cause mortality (see next section). In addition, however, we also performed additional projections, to illustrate the effects of the different steps in our projection approach (see the prefinal section of this Appendix). All our projections are based on 50,000 simulations, generated from the forecasts of the time series model. For future e0 we used the median values, and the 95% projection intervals.

## Lee–Carter mortality projection

The Lee–Carter (LC) model decomposes logged age-specific mortality rates over time [$\ln(m_{x,t})$] into an average age effect across time a(x), an overall time trend across all ages k(t), and an age-specific adjustment b(x) of this overall time trend (*Lee and Carter, 1992*). Projection into the future involves extrapolating the overall time trend k(t). The formula for the Lee–Carter model reads as:

$$ln\left(m_{x,t}\right) = a_x + b_x k_t + \varepsilon_{x,t} \tag{1}$$

We generated the parameter estimates based on Poisson-likelihood (*Brouhns et al., 2002*; *Renshaw and Haberman, 2006*). For the projection of the period parameter in the Lee–Carter model, we applied the common assumption of a random walk with drift ( = ARIMA(0,1,0) with drift) (*Lee and Carter, 1992*; *Renshaw and Haberman, 2006*).

## Steps involved in our projection

Our projection involved four steps.

1. Determining the long-term decline in mortality and life expectancy without the combined effect of smoking, obesity, and alcohol.
2. Projecting non-lifestyle-attributable mortality into the future, thereby taking into account the mortality experience of other populations.
3. Obtaining estimates of future mortality that is attributable to smoking, obesity, and alcohol.
4. Combining the projected non-lifestyle attributable mortality rates with the projected lifestyle-attributable mortality fractions.

### Step (1) Determining the long-term trend in mortality and life expectancy without the effect of smoking, obesity, and alcohol

As a first step in our projection, we assessed the trends over time in mortality and life expectancy without the effect of smoking, obesity, and alcohol. More specifically we assessed the gain in life expectancy at birth (e0) over the period 1990–2014 for non-lifestyle-attributable mortality and compared this with the observed gain for all-cause mortality. To this end, we used existing recent age- and sex-specific estimates of the share of mortality due to smoking, obesity and alcohol (*Janssen et al., 2021*) for ages 20–100, which we refer to as lifestyle-attributable mortality fractions (LAMF).

These age- and sex-specific lifestyle-attributable mortality fractions were estimated based on age- and sex-specific smoking-, obesity-, and alcohol-attributable mortality fractions from earlier studies (*Janssen, 2020*; *Janssen et al., 2020b*; *Janssen et al., 2020d*). Smoking-attributable mortality fractions (1950–2014; 35–100 M; 40–100 F) were indirectly estimated by applying a simplified version of the commonly applied Peto–Lopez methodology (*Peto et al., 1992*; *Janssen et al., 2013*;

*Janssen, 2020*) to lung cancer mortality data from the WHO Mortality Database (update April 11 2018) (*WHO, 2018b*), and to additional epidemiological information regarding smoking from the American Cancer Society's Cancer Prevention Study II (ACS-CPSII) (*Peto et al., 1992*; *Thun et al., 1997*). Obesity-attributable mortality fractions (1975–2016; 20–100) were estimated by applying the population-attributable fraction (PAF) formula to age- and sex-specific obesity prevalence data from the NCD Risk Factor Collaboration study 2017 (*Abarca-Gómez et al., 2017*) and age group- and sex-specific relative risks of dying from obesity from a meta-review (*DYNAMO-HIA Consortium, 2010*; *Janssen et al., 2020a*). Alcohol-attributable mortality fractions (1990–2016; 20–100) were esti-mated based on alcohol-attributable mortality rates obtained from the Global Burden of Disease Study 2017 (*Stanaway et al., 2018*; *GBD, 2018*). To ages 65+, we applied the age pattern observed for the main group of causes of death wholly attributable to alcohol (*Semyonova et al., 2014*), using data from the WHO Mortality Database (*WHO, 2018b*) and the *Human Cause-of-Death Database, 2017*. In the estimation of smoking-, obesity-, and alcohol-attributable mortality fractions, Loess smoothing was used to convert estimates by five-year age groups into estimates by single years of age. By setting the smoking-attributable mortality fractions for ages 20–34 (men) and ages 20–39 (women) to zero, and by restricting the data to the same time period, we ended up with estimated age- and sex-specific smoking-, obesity-, and alcohol-attributable mortality fractions for the 1990–2014 period for ages 20–100.

We, consequently, estimated the past age-and sex-specific lifestyle-attributable mortality frac-tions for the three lifestyles combined (LAMF) (1990-2014; 20-100), by multiplicatively aggregating the population-attributable mortality fractions for the individual risk factors ($\text{PAF}_i$) using the formula:

$$LAMF = PAF_{1.n} = 1 - \prod_{i=1}^{n}(1 - PAF_i) \tag{2}$$

(*Ezzati et al., 2003*; *Janssen et al., 2021*). For ages 0–20 we assumed these fractions to be zero.

By applying these age- and sex-specific lifestyle-attributable mortality fractions to age-and sex-specific all-cause mortality rates, we obtained age- and sex-specific non-lifestyle-attributable mortal-ity rates (1990–2014; 0–100).

To obtain the age- and sex-specific all-cause and non-lifestyle-attributable mortality rates for ages 0–130 to be used as input in our life table calculations, we obtained the respective mortality rates for ages 101–130 by applying the Kannisto model of old-age mortality (*Thatcher et al., 1998*) to mortality for ages 80–100 for all-cause mortality and non-lifestyle-attributable mortality, respectively.

## Step (2) Projecting non-lifestyle-attributable mortality into the future, thereby taking into account the mortality experience of other populations

To project age- and sex-specific non-lifestyle-attributable mortality rates (1990–2014; 0–100) into the future, thereby taking into account the mortality experience of other populations, we applied the often used Li and Lee coherent projection methodology (*Li and Lee, 2005*; *Stoeldraijer, 2019*). In doing so, we selected women in France, Spain and Italy as the forerunner populations in terms of life expectancy in Europe because they exhibit both very high recent life expectancy values and very favourable long-lasting past trends in life expectancy (measured by strong increases in e0 over the 1970–2011 period among non-Eastern European countries) (*Stoeldraijer, 2019*, p. 106). Our deci-sion to use forerunner populations as the reference populations is in line with the seminal work by James Vaupel (e.g., *Oeppen and Vaupel, 2002*), and our decision to use female populations is in line with the recent work by *Bergeron-Boucher et al., 2018*. In addition, using three populations instead of one is regarded more robust.

The Li–Lee (LL) coherent projection methodology, in essence, applies the Lee–Carter (LC) meth-odology twice. First, the LC model is applied to the forerunner populations, resulting in a common time trend (=a common improvement over time across all ages) K(t), a common age effect that adjusts the rate of mortality decline for each age B(x) and a common average age-specific death rate over time A(x). Second, the LC model is applied to the residuals, resulting in population-specific time trends k(t,i), population-specific age effects b(x,i), and population-specific average age-specific death rates a(x,i). These parameters are subsequently combined into separate models for each pop-ulation, by the formula below:

$$ln\left(m_{x,t,i}\right) = a_{x,i} + B_x K_t + b_{x,i}^{res} k_{t,i}^{res} + \varepsilon_{x,t,i} \tag{3}$$

Similar as to the Lee–Carter model, we generated the parameter estimates based on Poisson-likelihood (*Brouhns et al., 2002*; *Renshaw and Haberman, 2006*).

The past trend in the common time trend K(t) is extrapolated using the ARIMA(1,1,0) model with drift, which provided the best fit (=minimum corrected Akaike Information Criterion (AICc)) for both all-cause mortality and non-lifestyle-attributable mortality. For the remaining population-specific time trends k(t,i), we used a random walk with no drift, and thus assumed non-stationarity, which means that we used the last observed difference between the population and the common (instead of mean reverting). This assumption is in line with the results of the statistical test of it in most cases (see *Appendix 1—table 1*). For comparative purposes, we applied the assumption to all populations. Our assumption implies that the population-specific time trends k(t,i) will rely on the past trend in K(t) among the forerunner populations. However this will not always result in parallel age-specific mortality trends between the respective population and the forerunner populations, because of differences between populations in the average age-specific death rate a(x,i) and the age-specific adjustments of the overall time trend b(x,i).

**Appendix 1—table 1.** Test for stationarity of k(t,i) within the coherent Li-Lee mortality projections, for all-cause mortality and non-lifestyle-attributable mortality in which French, Italian, and Spanish women are selected as the forerunner populations, for the 18 European countries, by sex.

| | All-cause mortality | | Non-lifestyle-attributable mortality | |
|---|---|---|---|---|
| | **Men** | **Women** | **Men** | **Women** |
| Austria | 1 | 1 | 1 | 1 |
| Belgium | 1 | 1 | 1 | 1 |
| Czech Republic | 1 | 1 | 1 | 1 |
| Denmark | 1 | 1 | 1 | 1 |
| Finland | 1 | 1 | 1 | 1 |
| France | 0 | 1 | 1 | 1 |
| Germany | 1 | 1 | 1 | 0 |
| Greece | 1 | 1 | 1 | 1 |
| Hungary | 2 | 1 | 1 | 1 |
| Ireland | 1 | 1 | 1 | 1 |
| Italy | 1 | 0 | 1 | 0 |
| Netherlands | 1 | 1 | 1 | 0 |
| Norway | 1 | 1 | 1 | 1 |
| Poland | 1 | 1 | 1 | 1 |
| Slovenia | 2 | 1 | 1 | 1 |
| Sweden | 1 | 1 | 1 | 1 |
| Switzerland | 1 | 1 | 1 | 1 |
| United Kingdom | 1 | 1 | 1 | 1 |

0 = stationary time series; i.e., the time series does not exhibit a trend.

1 or 2 = non-stationary time series.

We obtained, for each sex-specific population, 50,000 simulated matrices of age (x) times period (t) of non-lifestyle-attributable mortality rates.

To obtain projected mortality rates for ages 101–130, for each sex-specific population, we extrapolated the age pattern for ages 80–100 up until 130 for each forecast year, by applying the Kannisto model of old-age mortality (*Thatcher et al., 1998*).

## Appendix 1—box 1.

Projections of smoking-, obesity-, and alcohol-attributable mortality (20–84).

For our estimate of future mortality that is attributable to smoking, obesity and alcohol, we used previously published data- and theory-driven projections of smoking-, obesity-, and alcohol-attributable mortality fractions (20–84) (*Janssen et al., 2013*; *Janssen et al., 2020b*; *Janssen et al., 2020d*). These projections follow the observed wave-shaped dynamic of the smoking epidemic (*Lopez et al., 1994*; *Janssen, 2020*) and the hypothesised wave shape of the obesity epidemic (*Xu and Lam, 2018*; *Jaacks et al., 2019*). For alcohol-attributable mortality, unrealistic future differences between countries were avoided by assuming that the current increases observed for selected countries will eventually turn into declines (*Janssen et al., 2020d*). All projections were based on the integration of these wave patterns into either a conventional age-period projection (for obesity prevalence, and consequently obesity-attributable mortality) or age-period-cohort projections (for smoking- and alcohol-attributable mortality). This was done by utilising the fact that a wave pattern occurs when the logistically transformed outcome measure has a quadratic shape.

The projections of obesity-attributable mortality were based on projections of obesity prevalence (*Janssen et al., 2020b*; *Janssen et al., 2020a*). For obesity prevalence, a wave shape was obtained by linearly extrapolating the speed of change ( = the first-order difference of the time trend) in the logit of prevalence, from a positive speed (increase) to a zero speed (maximum level of the quadratic shape) to a negative speed (decline). This is implemented by applying the age-period Lee–Carter mortality model to logistically transformed age-specific obesity prevalence data (1975–2016). Subsequently, the first-order difference of the general time trend $k(t)$ is linearly extrapolated by applying a second-order random walk with negative drift. This linear extrapolation is applied from 2000 onwards in the majority of countries, but from 1980 onwards for Central and Eastern European women.

Smoking-attributable mortality fractions (1950–2014/LAY) and alcohol-attributable mortality fractions (1990–2016/LAY) are projected by integrating the wave pattern of epidemics into age-period-cohort mortality modelling and projection (*Janssen et al., 2020c*; *Janssen et al., 2020d*). That is, the age-period-cohort model by *Cairns et al., 2009* is applied to the respective fractions using a generalised logit function as a link function. Subsequently, the (decelerating) increases in the period parameter (for smoking-attributable mortality among women, and for alcohol-attributable mortality in most Central and Eastern European countries and selected North-Western European countries) are projected by a quadratic curve with correlated errors to obtain future declines. Declines in the period parameter as well as the recent trend in the cohort parameter (after removing unreliable outer cohorts) are extrapolated using time-series forecasting (best ARIMA model under some restrictions).

In performing these projections, upper and lower bounds are implemented to ensure that the projections will not lead to unlikely crossovers between countries or between men and women, and will not lead to zero, in line with the theory (*Lopez et al., 1994*; *Xu and Lam, 2018*; *Jaacks et al., 2019*). More specifically, for obesity prevalence, the population- and age-specific prevalence in 1975 is assumed as the lower bound, except for Eastern European women (other than women in Estonia and Hungary), for whom the age-specific prevalence for men in 1975 is assumed. For alcohol-attributable mortality, different lower bounds by country group and sex were implemented based on their past trends and their past (peak) levels of age-standardised alcohol-attributable mortality fractions (AAMF). For the smoking-attributable mortality fractions (SAMF), a 5% smoking prevalence was assumed as the lower bound for both men and women. In addition, for women, the age-specific SAMF observed for Danish women in the year in which the all-age SAMF was highest is set as the upper bound.

## Step (3) Obtaining estimates of future mortality that is attributable to smoking, obesity and alcohol

We obtained projected lifestyle-attributable mortality fractions (0–130) based on previously published data- and theory-driven projections of smoking-, obesity-, and alcohol-attributable mortality (20–84) (*Janssen et al., 2020d*; *Janssen et al., 2020b*; *Janssen et al., 2020a*). See *Appendix 1—box 1* for some essential information on these projections.

We obtained the projected fractions by lifestyle factor and sex-specific population, by means of 50,000 simulation matrices of age (x) times period (t), for ages 20 up until 84, and for 2015 (or the first available year in the projection) up to 2065. See *Appendix 1—table 2* for the available projection years per country and lifestyle factor. The simulation matrices were generated from the forecasts of the different underlying time series models.

**Appendix 1—table 2.** Overview of the data used for the projections per lifestyle factor and combined.

| Age | Smoking-attributable mortality fractions (SAMF) 35–100 M; 40–100 F | | Alcohol-attributable mortality fractions (AAMF) 20–100 | | Obesity-attributable mortality fractions (OAMF)* 20–100 | | Lifestyle-attributable mortality fractions (LAMF)[†] 20–100[‡] | |
|---|---|---|---|---|---|---|---|---|
| Years by country | | | | | | | | |
| Austria | 1955 | 2014 | 1990 | 2014 | 1975 | 2016 | 1990 | 2014 |
| Belgium | 1954 | 2015 | 1990 | 2015 | 1975 | 2016 | 1990 | 2015 |
| Czech Republic | 1953 | 2016 | 1990 | 2016 | 1975 | 2016 | 1990 | 2016 |
| Denmark | 1951 | 2015 | 1990 | 2016 | 1975 | 2016 | 1990 | 2015 |
| Finland | 1952 | 2015 | 1990 | 2015 | 1975 | 2016 | 1990 | 2015 |
| France | 1950 | 2015 | 1990 | 2015 | 1975 | 2016 | 1990 | 2015 |
| Germany | 1970 | 2015 | 1990 | 2015 | 1975 | 2016 | 1990 | 2015 |
| Greece | 1961 | 2013 | 1990 | 2013 | 1981 | 2016 | 1990 | 2014[§] |
| Hungary | 1955 | 2014 | 1990 | 2014 | 1975 | 2016 | 1990 | 2014 |
| Ireland | 1950 | 2014 | 1990 | 2014 | 1975 | 2016 | 1990 | 2014 |
| Italy | 1951 | 2014 | 1990 | 2014 | 1975 | 2016 | 1990 | 2014 |
| Netherlands | 1950 | 2016 | 1990 | 2016 | 1975 | 2016 | 1990 | 2016 |
| Norway | 1951 | 2014 | 1990 | 2014 | 1975 | 2016 | 1990 | 2014 |
| Poland | 1959 | 2014 | 1990 | 2014 | 1975 | 2016 | 1990 | 2014 |
| Slovenia | 1985 | 2014 | 1990 | 2014 | 1983 | 2016 | 1990 | 2014 |
| Sweden | 1951 | 2016 | 1990 | 2016 | 1975 | 2016 | 1990 | 2016 |
| Switzerland | 1951 | 2015 | 1990 | 2016 | 1975 | 2016 | 1990 | 2015 |
| United Kingdom | 1950 | 2015 | 1990 | 2016 | 1975 | 2016 | 1990 | 2015 |

*We projected obesity prevalence data (1975–2016) into the future. To obtain the OAMF, we only needed the RRs of dying from obesity. However, to obtain age-standardised fractions, we also needed all-cause mortality data.

[†]See projection step three for details on how we dealt with projections not being available for all the three lifestyle factors for the years 2015–2016.

[‡]By setting the SAMF for ages 20–34 (M) and ages 20–39 (F) to zero.

[§]We implemented the median projected AAMF and SAMF for Greece as the observed AAMF and SAMF in 2014 so that we could calculate the lifestyle-attributable mortality up until 2014, and so that we could apply it to the all-cause mortality rates to obtain non-lifestyle-attributable mortality rates. For this purpose, we used all-cause mortality from HMD from 26 February 2020.

Subsequently, we combined the projections (50,000 simulations) of the smoking-, obesity-, and alcohol-attributable mortality fractions to obtain 50,000 simulations of lifestyle-attributable mortality

fractions (LAMF) (20–84) by again aggregating them multiplicatively (see *Equation 2*). That is, for b = 1, ..., 50,000 simulations we applied *Equation 2* to the matrices SAMF_x,t[b], OAMF_x,t[b], and AAMF_x,t[b].

Because the projections for smoking-, obesity-, and alcohol-attributable mortality are not available for the same years (for obesity from 2017 onwards and for smoking and alcohol in some cases only from 2015 onwards) (see *Appendix 1—table 1*), we adopted a slightly different procedure for the years 2015 and 2016. That is, for these years, we combined the available observed values for the specific lifestyle factor(s) with the projected values (50,000 simulations) for the specific lifestyle factor(s) for which we already had projections. In practice, we did so by turning the observed values into matrices so that we obtained equally large matrices everywhere.

As a next step, we extrapolated each of the 50,000 simulated age patterns of LAMF (20–84) to age 100 by applying Loess smoothing (for each year-simulation combination separately). More specifically, we applied Loess smoothing with degree = 2 and span = 0.75 on log(LAMF) for ages 20–84 and 130, where the LAMF at age 130 is set to 0.000001 so that a decline is enforced. To obtain LAMF for ages 85–100, we extrapolated using the fitted Loess curve, and subsequently took the exponent. This generated 50,000 simulated matrices of LAMF_x,t for x = 20–100 and t. For ages 0–19 and 101–130, we set the LAMF_x,t to zero.

To illustrate our projection of lifestyle-attributable mortality fractions, we calculated past and future all-age estimates of lifestyle-attributable mortality fractions. To avoid dependence on future death or population numbers, we calculated age-standardised mortality fractions by applying the sex- and country-specific age composition of all-cause mortality in 2010. The age-standardised lifestyle-attributable mortality fractions were calculated by directly applying the multiplicative aggregation to the past and future (median) age-standardised smoking-, obesity-, and alcohol-attributable mortality (ages 20–84).

## Step (4) Combining the projected non-lifestyle attributable mortality rates with the projected lifestyle-attributable mortality fractions

In the final step of our projection approach, we combined the projected non-lifestyle mortality rates with the projected lifestyle-attributable mortality fraction to obtain our projected all-cause mortality rates. For this purpose, we extended the approach that *Janssen et al., 2013* developed for combining the projection of non-smoking-attributable mortality with smoking-attributable mortality fractions.

We ended up with the following formula:

$$m_{x,t}^{allcause} = m_{x,t}^{non-lifestyle} \cdot \left( \frac{1}{1 - LAMF_{x,t}} \right) \tag{4}$$

Where $m_{x,t}^{allcause}$ stands for age- and year-specific all-cause mortality rates, $m_{x,t}^{non-lifestyle}$ stands for age- and year-specific non-lifestyle-related mortality rates, and $LAMF_{x,t}$ stands for the age- and year-specific lifestyle-attributable mortality fraction.

We applied this formula to the 50,000 simulation matrices of coherently projected non-lifestyle-attributable mortality rates (0–130) and the 50,000 simulation matrices of projected LAMF (0–130) to obtain the 50,000 simulation matrices for all-cause-mortality based on our projection.

## Additional projections

As mentioned, we compared our projection outcomes foremost with the outcomes of a Lee–Carter extrapolation (*Lee and Carter, 1992*) applied to all-cause mortality. In addition, however, we also performed additional projections, to illustrate the effects of the different steps in our projection approach. That is, we also performed a Lee–Carter projection of non-lifestyle-attributable mortality, a Li–Lee coherent projection of all-cause mortality, and a Li–Lee coherent projection of non-lifestyle-attributable mortality. In addition, we assessed the effect of purely incorporating lifestyle, without employing a coherent mortality projection, which constitutes of applying the Lee–Carter projection to non-lifestyle-attributable mortality and adding the estimated future lifestyle-attributable mortality. For these projections, we adopted similar specifications as for our projection approach, and as for

the Lee–Carter extrapolation of all-cause mortality. *Supplementary file 1C* compares e0 in 2065 for all the different projections we employed.

## Software

We used the R software for all elements of our analysis and projection. We implemented the Lee–Carter and Li–Lee models using the StMoMo package in R (*Millossovich, 2018*).

