## [Decision Letter]

**Acceptance summary:**

Combining a few analytical strategies, this manuscript proposes a new approach to forecast mortality that distinguishes future changes in lifestyle- and non-lifestyle attributable mortality on life expectancy projection. The authors use data from 18 European countries to test their model and the results yield more optimistic forecasts than other well-established forecasting methods; future generations will have an increased life expectancy than previously expected. The proposed methodology could bring significant benefits when applied to other contexts and represents an important contribution to the field of life expectancy forecasting.

**Decision letter after peer review:**

Thank you for submitting your article "Future life expectancy in Europe taking into account the impact of smoking, obesity and alcohol" for consideration by *eLife*. Your article has been reviewed by 2 peer reviewers, and the evaluation has been overseen by a Reviewing Editor and a Senior Editor. The following individuals involved in review of your submission have agreed to reveal their identity: Ugofilippo Basellini (Reviewer #1); Collin Payne (Reviewer #2).

As is customary in *eLife*, the reviewers have discussed their critiques with one another. What follows below is the Reviewing Editor's edited compilation of the essential and ancillary points provided by reviewers in their critiques and in their interaction post-review. Please submit a revised version that addresses these concerns directly. Although we expect that you will address these comments in your response letter, we also need to see the corresponding revision clearly marked in the text of the manuscript. Some of the reviewers' comments may seem to be simple queries or challenges that do not prompt revisions to the text. Please keep in mind, however, that readers may have the same perspective as the reviewers. Therefore, it is essential that you attempt to amend or expand the text to clarify the narrative accordingly.

Essential Revisions (for the authors):

1. Line 206: "France, Spain, and Italy". I am unsure why you included Spain in your forerunner population, since you do not have Spain in your results. This seems not very consistent, and it should be better justified and clarified. I suppose that you do not have lifestyle-attributable mortality for Spain? How do your results change if you exclude Spain from the forerunners (I imagine very little)?

2. Line 392-393: "However, in our view, the added value of integrating lifestyle into mortality projections outweighs the uncertainties that come with it." This should, and can, be empirically tested with out-of-sample exercises.

3. SM Methodology: "All our projections are based on 50,000 simulations" and "We obtained, for each sex-specific population, 50,000 simulated matrices". I was not able to understand how you performed these simulations. Are these generated from the forecasts of the time series model? Or do you employ a bootstrap on the deviance residuals of the fitted models? Or both? More details on how you compute variability of your forecasts are needed.

4. The following text is an expanded explanation of the Public Evaluation of Reviewer 2: "My main issue with the paper as written concerns the assumptions built into the Li-Lee projection of non-lifestyle attributable mortality. I wonder how sensitive the results are to the choice of reference populations here--that is, I'm not sure that female populations of France, Spain, and Italy are the best populations to use as reference here.

These populations currently have high life expectancies, but also have histories of substantial mortality shocks and had much higher mortality rates in the near past. My issue here comes down to the essential difference between current mortality rates and current mortality conditions. The low age-specific mortality rates of these populations are a function of both current mortality conditions (e.g. period effects/current conditions in terms of development, medical care, etc) and of the history of cohort mortality selection pressures among those currently in the population. The chosen reference populations all have older cohorts who have experienced fairly severe mortality selection in early life, which may lead to smaller groups of more selected, "hardier" individuals surviving to later life. I worry that choosing these populations may provide an over-optimistic reference point for convergence (indeed, these three countries have among the largest gaps between period e0 and currently attained cohort e0--see https://doi.org/10.1080/00324728.2019.1618480). So I'd be interested in seeing whether results differed by using a set of low-mortality countries that have faced less severe life-course mortality pressures--e.g. Sweden, Norway, and Switzerland. My hunch is that your life expectancy projections will decline a little, though not massively. But I think it's worth pushing on this assumption a bit to see how sensitive it is."

*Reviewer #1 (Recommendations for the authors):*

The manuscript introduces a new approach to forecast mortality that explicitly considers the role of lifestyle epidemics (smoking, obesity and alcohol) on the dynamics of mortality. Specifically, a four-step projection methodology is proposed, which distinguishes between non-lifestyle-attributable and lifestyle-attributable mortality. The former is forecast using an extrapolative model, the latter is forecast using a data- and theory-driven approach, and the two components are combined to obtain all-cause mortality forecasts.

The model is applied to 18 European countries and compared to three other well-established forecasting methods (Lee-Carter, United Nations and Eurostat). The proposed approach results in more optimistic forecasts than the other methods, suggesting that future individuals will live longer lives than previously expected. Moreover, forecasts of the proposed methodology are more realistic as they do not display implausible crossovers between sexes and countries which characterize the other methods.

For these reasons, the proposed methodology appears to be an important contribution to the mortality forecasting literature. However, the manuscript has some weaknesses that the authors should address to further improve their work.

Weaknesses

While the forecasts of the proposed approach appear to be more realistic than those of the benchmark model of Lee and Carter (1992), the forecast accuracy of the proposed model is not empirically evaluated. In the recent mortality forecasting literature, great attention has been devoted to out-of-sample validation exercises, which are employed to measure the accuracy of the forecasts (see, e.g.,Shang et al. 2011,Bergeron-Boucher et al. 2017). A point and interval forecast accuracy evaluation of the proposed model as compared to the Lee-Carter one would provide concrete evidence of the new model's strengths. Clearly, given the limited time series available to the authors, only a short-term forecast evaluation can be performed (e.g. 5 and 10 years forecasts).

Moreover, the results and discussions of the authors are exclusively based on life expectancy, which is certainly a very important summary measure of mortality, but it does not capture everything. No results are shown or discussed in terms of forecast mortality rates and lifespan inequality. The former are very relevant to understand the plausibility of the forecast mortality age-pattern. The latter is another important summary measure of mortality that complements life expectancy (van Raalte et al., 2018), and it should be considered in mortality forecasting (Bohk-Ewald et al., 2017).

The data employed to fit the model is limited to 25 years (1990--2014) because lifestyle-attributable mortality data is available only for this period of time. Based on this (rather short) fitting period, the authors forecast mortality for a period of time twice as long (up to 2065, that is a 51 year forecast). This seems rather unbalanced. Moreover, forecasts of lifestyle-attributable mortality seem to be driven more by theory/expert opinion rather that by observed time series, especially for females. I think these potential limitations should be more clearly specified in the manuscript.

Finally, the principal added value of the manuscript – the new approach to forecast mortality – is currently not available to the general public: "Because of the large size of the many different underlying simulation matrices, we are restricted in sharing these data. … The R codes can be requested from the author." One simple way to address this problem is to share a much smaller set of simulations (say 100 instead of 50,000) and warn the users to increase the number of simulations when running the codes on their machines. This would directly provide the research community with the routines to employ the proposed methodology.

References

Bergeron-Boucher, M.-P., V. Canudas-Romo, J. Oeppen, and J. W. Vaupel (2017). Coherent forecasts of mortality with compositional data analysis. Demographic Research 37, 527-566.

Bohk-Ewald, C., M. Ebeling, and R. Rau (2017). Lifespan Disparity as an Additional Indicator for Evaluating Mortality Forecasts. Demography 54 (4), 1559-1577.

Lee, R. D. and L. R. Carter (1992). Modeling and forecasting US mortality. Journal of the American Statistical Association 87 (419), 659-671.

Shang, H. L., H. Booth, and R. Hyndman (2011). Point and interval forecasts of mortality rates and life expectancy: A comparison of ten principal component methods. Demographic Research 25, 173-214.

van Raalte, A. A., I. Sasson, and P. Martikainen (2018). The case for monitoring life-span inequality. Science 362 (6418), 1002-1004.

*Reviewer #2 (Recommendations for the authors):*

The authors have conducted an ambitious projection exercise, drawing together a number of methods and approaches to seek to understand the impacts of future changes in lifestyle- and non-lifestyle attributable mortality on prospects for life expectancy increase in a set of European countries. This focus on the underlying dynamics of mortality change--that is, disaggregating the long-term changes in non-lifestyle attributable mortality patterns from the much more variable lifestyle-attributable mortality patterns--represents a real step forward in life expectancy projection modeling. These methods could have considerable benefits for projections in other contexts.

My main issue with the paper as written concerns the assumptions built into the Li-Lee projection of non-lifestyle attributable mortality, where I would encourage the authors to test the sensitivity of their projections to alternative choices of reference countries (looking beyond Italy, Spain, and France).

---

## [Author Response]

Essential Revisions (for the authors):1. Line 206: "France, Spain, and Italy". I am unsure why you included Spain in your forerunner population, since you do not have Spain in your results. This seems not very consistent, and it should be better justified and clarified. I suppose that you do not have lifestyle-attributable mortality for Spain? How do your results change if you exclude Spain from the forerunners (I imagine very little)?

We selected women in France, Spain, and Italy as the forerunner populations in terms of life expectancy in Europe because they exhibit both very high recent life expectancy values and very favourable long-lasting past trends in e0 (measured by strong increases in e0 over the 1970-2011 period among non-Eastern European countries) (Stoeldraijer 2019) (see page 4 of the Supplementary Data and Methods (now Appendix 1)). Our decision to use forerunner populations as the reference populations is in line with the seminal work by James Vaupel (e.g., Oeppen and Vaupel 2002), and our decision to use female populations is in line with the recent work by Bergeron-Boucher et al. 2018. In addition, using three populations instead of one is regarded more robust.

While we had lifestyle-attributable mortality data for Spain, we ended up excluding the projection outcomes for Spain because we were not able to obtain realistic long-term projections of smoking-attributable mortality for Spanish women. Specifically, we found that the projections of smoking-attributable mortality for Spanish women were not robust to the selection of different historical periods (see Janssen et al. 2020 Tobacco Control). Nonetheless, our main argument for why we selected women in France, Spain, and Italy as the forerunner populations in Europe in terms of life expectancy holds.

Given that the gain in e0 over the 1990-2014 period was slightly higher for Spanish women than it was for women in Italy and France (see Table 1), but that the population size was smaller in Spain than in Italy and France, the exclusion of Spanish women from the forerunner populations will most likely result in only slightly lower future life expectancy values. This expectation proves indeed in line with the results of our sensitivity analysis regarding the selection of the forerunner populations that we present in Author response table 1. That is, the coherent projection of non-lifestyle-attributable mortality in which we did not include Spanish women as forerunner population resulted in life expectancy values in 2065 that were – on average – 0.21 years lower for men, and 0.14 years lower for women. For all-cause mortality we see a similar effect (slightly larger difference).

**Author response table 1. resptable1:** Effect of the inclusion of alternative reference populations within coherent mortality projections* on the coherently projected e0 values in 2065 for all-cause mortality and for non-lifestyle-attributable mortality.

	All-cause mortality	Non-lifestyle attributable mortality
Country	Coh. Li-Lee: sensivity I	Coh. Li-Lee: sensivity II	Coh. Li-Lee: sensivity III	Coh. Li-Lee: sensivity I	Coh. Li-Lee: sensivity II	Coh. Li-Lee: sensivity III
**Men**						
Austria	-0.20	-2.19	-0.76	-0.13	-0.77	-0.94
Belgium	-0.17	-2.08	-0.61	-0.16	-0.81	-0.90
Czech Republic	-0.24	-2.33	-0.71	-0.23	-0.76	-1.09
Denmark	-0.17	-2.11	-0.60	-0.14	-0.71	-0.91
Finland	-0.26	-2.13	-0.83	-0.22	-0.73	-1.00
France	-0.21	-2.12	-0.75	-0.09	-0.78	-0.78
Germany	-0.23	-2.09	-0.85	-0.22	-0.70	-0.95
Greece	-0.27	-2.05	-0.80	-0.31	-0.82	-1.01
Hungary	-0.19	-2.11	-0.65	-0.24	-0.83	-1.14
Ireland	-0.17	-2.05	-0.80	-0.13	-0.82	-1.06
Italy	-0.28	-1.86	-0.74	-0.18	-0.73	-0.92
Netherlands	-0.16	-1.94	-0.67	-0.17	-0.76	-0.84
Norway	-0.35	-1.83	-0.77	-0.31	-0.82	-1.03
Poland	-0.22	-2.14	-0.67	-0.28	-0.79	-1.01
Slovenia	-0.22	-2.03	-0.68	-0.15	-0.67	-1.00
Sweden	-0.27	-1.79	-0.71	-0.29	-0.64	-0.89
Switzerland	-0.23	-1.82	-0.67	-0.19	-0.69	-0.89
United Kingdom	-0.26	-2.02	-0.70	-0.23	-0.80	-0.91
*Average*	*-0.23*	*-2.04*	*-0.72*	*-0.21*	*-0.76*	*-0.96*
**Women**						
Austria	-0.19	-1.61	-0.56	-0.16	-0.73	-0.81
Belgium	-0.21	-1.67	-0.72	-0.09	-0.76	-0.83
Czech Republic	-0.24	-1.70	-0.64	-0.10	-0.68	-0.87
Denmark	-0.19	-1.77	-0.66	-0.08	-0.53	-0.58
Finland	-0.22	-1.69	-0.69	-0.14	-0.73	-0.86
France	-0.23	-1.37	-0.54	-0.12	-0.71	-0.77
Germany	-0.21	-1.55	-0.66	-0.16	-0.66	-0.82
Greece	-0.27	-1.52	-0.72	-0.13	-0.69	-0.85
Hungary	-0.18	-1.79	-0.65	-0.16	-0.79	-1.03
Ireland	-0.28	-1.83	-0.77	-0.18	-0.81	-0.80
Italy	-0.25	-1.59	-0.65	-0.16	-0.70	-0.85
Netherlands	-0.25	-1.57	-0.68	-0.15	-0.66	-0.75
Norway	-0.24	-1.67	-0.66	-0.09	-0.66	-0.85
Poland	-0.18	-1.85	-0.69	-0.19	-0.71	-0.90
Slovenia	-0.20	-1.62	-0.68	-0.14	-0.60	-0.85
Sweden	-0.23	-1.56	-0.65	-0.14	-0.59	-0.74
Switzerland	-0.19	-1.45	-0.59	-0.16	-0.66	-0.73
United Kingdom	-0.23	-1.65	-0.68	-0.18	-0.56	-0.70
*Average*	*-0.22*	*-1.64*	*-0.66*	*-0.14*	*-0.68*	*-0.81*

*original = using women in France, Spain and Italy as the reference population.sensitivity 1 = using only women in France and Italy as the reference population.

sensitivity 2 = using women in Sweden, Norway and Switzerland as the reference population.sensitivity 3 = using the average across the 18 countries included in the analysis as the reference population.

Note: the sensitivity analyses were all based on 1000 simulations, and we assumed non-stationarity for the remaining population-specific time trends k(t,i).

In our revised manuscript, we now provide some additional details regarding why we selected women in France, Spain, and Italy as the forerunner populations in Europe (Results section; lines 219-222; Appendix 1 – Supplementary Data and Methods – page 4). We have also included a discussion regarding the selection of forerunner populations in our discussion (see as well our reply to essential revision nr. 4).

2. Line 392-393: "However, in our view, the added value of integrating lifestyle into mortality projections outweighs the uncertainties that come with it." This should, and can, be empirically tested with out-of-sample exercises.

Although out-of-sample exercises can indeed be used in assessing the accuracy of forecasts, we do not think that such an exercise would be able to test whether the added value of integrating lifestyle into mortality projections outweighs the uncertainties that come with doing so. That is, as well as the accuracy of forecasts, which out-of-sample exercises can indeed help to determine, there are many additional evaluation criteria for assessing the performance of forecasts (see Cairns et al. 2009). These include, for example, biological reasonableness, the plausibility of outcomes, and the robustness of the forecast outcomes (Cairns et al. 2009). Because these latter issues are more difficult to quantify, they are often ignored. Instead, accuracy alone is tested by means of out-of-sample (=backtesting) exercises. We believe that our mortality projection approach is particularly promising because its outcomes can be considered more realistic, more robust, and more insightful than previous mortality projections that were purely based on extrapolation.

These points are highlighted in the remainder of the respective paragraph:

“However, in our view, the added value of integrating lifestyle into mortality projections outweighs the uncertainties that come with it. First, it provides a better approximation of the underlying mortality trend. Second, our projection is driven not only by data, but by theory, and is preceded by a careful study of past trends. Third, by distinguishing between the underlying mortality trend and the additional effect of the three lifestyle factors combined, these projections provide much greater transparency and insight than a mere mechanical extrapolation of past trends could.”

Moreover, in response to the comment by reviewer 1 on this issue, we have now added at the end of this paragraph an explicit reference to the different evaluation criteria used for mortality projections, and have also added a reference to our more realistic outcomes (lines 438-446):

“Fourth, our approach resulted in more realistic differences between countries and sexes than those projected by other approaches (see the next section as well). The above mentioned four added values of our mortality projection approach can be clearly linked to important criteria for evaluating the performance of mortality forecasts in addition to accuracy: namely, robustness, plausibility, reasonableness, and transparency (Cairns et al. 2009). Moreover, the outcomes of our projection approach – which rely above all on the coherent projection of non-lifestyle-attributable mortality – are less dependent on the explicit choices that underlie general mortality forecasts (e.g., the choice of the forerunner populations and the historical time period)”.

In addition, in our text, we avoided using the words “accurate” and “inaccurate” as much as possible and replaced these terms with “reliable” and “unreliable”.

3. SM Methodology: "All our projections are based on 50,000 simulations" and "We obtained, for each sex-specific population, 50,000 simulated matrices". I was not able to understand how you performed these simulations. Are these generated from the forecasts of the time series model? Or do you employ a bootstrap on the deviance residuals of the fitted models? Or both? More details on how you compute variability of your forecasts are needed.

Yes, the simulations are generated from the forecasts of the time series model. We did not employ a bootstrap on the deviance residuals of the fitted models. Thus, we did not take into account parameter uncertainty. We have added these details to the Supplementary Data and Methods (page 2 and page 5)(=now Appendix 1).

4. The following text is an expanded explanation of the Public Evaluation of Reviewer 2: "My main issue with the paper as written concerns the assumptions built into the Li-Lee projection of non-lifestyle attributable mortality. I wonder how sensitive the results are to the choice of reference populations here--that is, I'm not sure that female populations of France, Spain, and Italy are the best populations to use as reference here.These populations currently have high life expectancies, but also have histories of substantial mortality shocks and had much higher mortality rates in the near past. My issue here comes down to the essential difference between current mortality rates and current mortality conditions. The low age-specific mortality rates of these populations are a function of both current mortality conditions (e.g. period effects/current conditions in terms of development, medical care, etc) and of the history of cohort mortality selection pressures among those currently in the population. The chosen reference populations all have older cohorts who have experienced fairly severe mortality selection in early life, which may lead to smaller groups of more selected, "hardier" individuals surviving to later life. I worry that choosing these populations may provide an over-optimistic reference point for convergence (indeed, these three countries have among the largest gaps between period e0 and currently attained cohort e0--see https://doi.org/10.1080/00324728.2019.1618480). So I'd be interested in seeing whether results differed by using a set of low-mortality countries that have faced less severe life-course mortality pressures--e.g. Sweden, Norway, and Switzerland. My hunch is that your life expectancy projections will decline a little, though not massively. But I think it's worth pushing on this assumption a bit to see how sensitive it is."

Yes, the choice of the reference or forerunner populations is indeed an important aspect of multi-population or coherent mortality forecasting (Stoeldraijer 2019; Kjærgaard et al. 2016). However, given that the trend in non-lifestyle-attributable mortality is more equal between countries and sexes than the trend in all-cause mortality (see Table 1), this choice will likely have a larger impact when coherently forecasting all-cause mortality than when coherently forecasting non-lifestyle-attributable mortality (see also Janssen and Kunst 2007 and Janssen et al. 2013). A comparison of the average (unweighted) gains in e0 over the 1990-2014 period for women in France, Italy, and Spain (4.8 for all-cause mortality; 5.0 for non-lifestyle-attributable mortality) with the average (unweighted) gains in e0 for women in Sweden, Norway, and Switzerland (4.1 for all-cause mortality; 4.8 for non-lifestyle-attributable mortality) shows that the selection of these latter populations as the forerunner populations would likely result in lower projection outcomes, particularly for all-cause mortality, and to a much lesser extent for non-lifestyle-attributable mortality. This expectation proves indeed in line with the results of our sensitivity analysis regarding the selection of the forerunner populations that we present in Author response table 1.

In response to the remark by the reviewer, we have now added a discussion on the choice of the reference populations in the “Appraisal of our projection methodology” section (lines 384 – 396): “However, as well as the potential effects on the outcomes of the exact coherent methodology that is applied (Stoeldraijer, 2019), it is important to take into account the choice of the reference population (Stoeldraijer 2019; Booth 2020). Using forerunner populations as the reference population, as advocated by among others Oeppen and Vaupel 2002 and Bergeron-Boucher et al. 2018, instead of an average across many populations (Li and Lee 2005; Eurostat 2020) will generally result in higher future life expectancy values (see the next section as well). However, given that the past trend in e0 for non-lifestyle-attributable mortality is more equal between countries and sexes than the past trend in e0 for all-cause mortality (see Table 1), which reference populations are chosen will likely have a larger impact when coherently forecasting all-cause mortality than when coherently forecasting non-lifestyle-attributable mortality (see also Janssen and Kunst 2007 and Janssen et al. 2013)”.

Reviewer #1 (Recommendations for the authors):The manuscript introduces a new approach to forecast mortality that explicitly considers the role of lifestyle epidemics (smoking, obesity and alcohol) on the dynamics of mortality. Specifically, a four-step projection methodology is proposed, which distinguishes between non-lifestyle-attributable and lifestyle-attributable mortality. The former is forecast using an extrapolative model, the latter is forecast using a data- and theory-driven approach, and the two components are combined to obtain all-cause mortality forecasts.The model is applied to 18 European countries and compared to three other well-established forecasting methods (Lee-Carter, United Nations and Eurostat). The proposed approach results in more optimistic forecasts than the other methods, suggesting that future individuals will live longer lives than previously expected. Moreover, forecasts of the proposed methodology are more realistic as they do not display implausible crossovers between sexes and countries which characterize the other methods. For these reasons, the proposed methodology appears to be an important contribution to the mortality forecasting literature. However, the manuscript has some weaknesses that the authors should address to further improve their work.WeaknessesWhile the forecasts of the proposed approach appear to be more realistic than those of the benchmark model of Lee and Carter (1992), the forecast accuracy of the proposed model is not empirically evaluated. In the recent mortality forecasting literature, great attention has been devoted to out-of-sample validation exercises, which are employed to measure the accuracy of the forecasts (see, e.g.,Shang et al., 2011,Bergeron-Boucher et al. 2017). A point and interval forecast accuracy evaluation of the proposed model as compared to the Lee-Carter one would provide concrete evidence of the new model's strengths. Clearly, given the limited time series available to the authors, only a short-term forecast evaluation can be performed (e.g. 5 and 10 years forecasts).

It is indeed the case that we did not empirically evaluate the forecast accuracy of the proposed model. There are three reasons why we made this choice. First, accuracy is only one of the many criteria for assessing the performance of forecasts (see Cairns et al. 2009). Among the other evaluation criteria are biological reasonableness, the plausibility of outcomes, the robustness of the forecast outcomes, and transparency (Cairns et al. 2009). Because these latter issues are more difficult to quantify, they are often ignored. Instead, only accuracy is tested by means of out-of-sample exercises (=backtesting). However, as we indicated in our discussion, we consider our mortality projection approach to be particularly promising because its outcomes can be considered more realistic, more robust, more theoretically informed, and more transparent than the outcomes of other approaches. Second, given the limited time series for lifestyle-attributable morality, we could only perform a short-term forecast evaluation, whereas our main aim is to produce reliable long-term forecasts. Third, given the profile of the readers of eLife, we believe that applying additional technical exercises would not necessarily be an improvement.

Related to our choice to avoid empirically evaluating the forecast accuracy of our model, we have avoided making any statements in our discussion that would suggest that our model is more accurate than the Lee-Carter model. Moreover, we now avoid the use of “accurate” and “inaccurate” as much as possible in the remainder of our text. In addition, we now explicitly refer to the different criteria for evaluating mortality projections in the last paragraph of the “Appraisal of our projection methodology” section.

Moreover, the results and discussions of the authors are exclusively based on life expectancy, which is certainly a very important summary measure of mortality, but it does not capture everything. No results are shown or discussed in terms of forecast mortality rates and lifespan inequality. The former are very relevant to understand the plausibility of the forecast mortality age-pattern. The latter is another important summary measure of mortality that complements life expectancy (van Raalte et al. 2018), and it should be considered in mortality forecasting (Bohk-Ewald et al., 2017).

We completely agree that in addition to life expectancy, other outcome measures of mortality forecasts are relevant and interesting. However, we decided to focus on life expectancy at birth in our manuscript because it is a very common summary measure of health; it is the most common output measure of mortality forecasts; it is easier to grasp than lifespan variation; and it summarizes the age pattern of mortality using a single measure. Given the more general audience of eLife, we do not consider it particularly helpful to include additional outcome measures as part of our submission. However, in response to the comment of the reviewer, we have clarified why we have chosen to focus on life expectancy, thereby referring to other outcome measures that are important as well (see lines 168 – 170; see Supplementary Data and Methods (now Appendix 1) page 2 – first paragraph).

In addition, we have added figures with the projected age-specific mortality rates (Author response image 1 and Author response image 2) and a table (Author response table 2) with the observed (2014) and projected (2065) standard deviation (SD) of the age-at-death distribution from age zero onwards ( = the dx column of the lifetable). This measure is often used to denote lifespan variation (see Edwards and Tuljapurkar, 2005; Garcia and Aburto, 2019).

Furthermore, we will make available the projected age-specific mortality rates (median and 95% and 90% projection intervals) on Open Science Framework (OSF).

**Author response image 1. respfig1:** Age-specific mortality pattern in 1990*, 2014*, and for different projections in 2065, by country, ages 0-130, men. * based on the adjusted rates up to 130 (because of zeros in the observed rates).

**Author response image 2. respfig2:** Age-specific mortality pattern in 1990*, 2014*, and for different projections in 2065, by country, ages 0-130, women. * based on the adjusted rates up to 130 (because of zeros in the observed rates).

**Author response table 2. resptable2:** Observed (2014) and projected (2065) lifespan variation from age zero onwards, by country and sex, according to our projection methodology, which takes into account the impact of smoking, obesity, and alcohol and the mortality experiences of forerunner countries (“lifestyle and coherent”); the benchmark Lee-Carter extrapolative mortality projection applied to all-cause mortality (“Lee-Carter”); and when purely accounting for smoking, obesity, and alcohol (“adding lifestyle”).

	MEN	WOMEN
		Projected SD* 2065		Projected SD* 2065
Country	SD* 2014	Lee-Carter	Adding lifestyle	Lifestyle and coherent	SD* 2014	Lee-Carter	Adding lifestyle	Lifestyle and coherent
Austria	14.33	10.88	11.11	10.31	13.05	9.69	8.73	8.91
Belgium	14.70	11.57	11.27	9.78	13.36	11.09	9.53	9.18
Czechia	14.36	10.96	12.02	10.69	12.61	9.23	8.53	9.22
Denmark	14.25	10.51	10.37	10.09	13.26	10.15	10.10	9.27
Finland	14.69	11.61	11.18	10.35	12.66	10.90	9.65	9.03
France	15.32	11.75	11.35	10.04	13.59	11.11	9.92	9.27
Germany	14.25	10.66	11.31	10.39	12.93	9.61	8.85	9.23
Greece	15.10	14.31	14.26	10.56	12.67	10.20	9.98	9.25
Hungary	15.11	12.77	11.23	11.21	13.90	11.38	9.33	10.01
Ireland	14.52	11.43	10.84	9.78	13.18	10.59	9.98	9.25
Italy	13.53	9.11	10.95	9.76	12.41	9.06	9.01	8.94
Netherlands	13.62	9.82	10.16	9.60	13.19	11.12	9.68	8.93
Norway	13.74	10.17	9.93	9.68	12.50	10.22	9.39	8.79
Poland	15.93	13.49	12.52	11.50	13.92	11.08	9.65	9.86
Slovenia	13.89	10.05	10.93	10.33	12.28	9.00	8.39	8.96
Sweden	13.60	10.31	10.13	9.47	12.49	9.88	9.15	8.69
Switzerland	13.84	9.45	10.29	9.81	12.59	9.37	8.79	8.66
United Kingdom	14.64	12.24	11.87	10.11	13.62	11.06	10.58	9.37
*Average*	*14.41*	*11.17*	*11.21*	*10.19*	*13.01*	*10.26*	*9.40*	*9.16*
*Min*	*13.53*	*9.11*	*9.93*	*9.47*	*12.28*	*9.00*	*8.39*	*8.66*
*Max*	*15.93*	*14.31*	*14.26*	*11.50*	*13.92*	*11.38*	*10.58*	*10.01*
*Variance*	*0.42*	*1.81*	*0.99*	*0.28*	*0.25*	*0.61*	*0.34*	*0.12*

* SD = Standard deviation of the age at death distribution (age 0-100).

The data employed to fit the model is limited to 25 years (1990-2014) because lifestyle-attributable mortality data is available only for this period of time. Based on this (rather short) fitting period, the authors forecast mortality for a period of time twice as long (up to 2065, that is a 51 year forecast). This seems rather unbalanced. Moreover, forecasts of lifestyle-attributable mortality seem to be driven more by theory/expert opinion rather that by observed time series, especially for females. I think these potential limitations should be more clearly specified in the manuscript.

In general, it is indeed considered optimal to base projections – and, in particular, extrapolations – on a historical period of about 40-50 years when projecting 40-50 years into the future (Technical Panel on Assumptions and Methods, 2003). Thus, it is a pity that because data on alcohol-attributable mortality were only available from 1990 onwards, we could base our forecast only on data from 1990 to 2014/LAY. However, we decided against restricting our projection horizon to 25 years in the future because we considered it essential to show that our approach is able to generate reliable predictions for the long-term future. It should also be noted that the use of a relatively long historical period stems from the desire to select the most reliable long-term past trend on which the extrapolations could be based. However, because of the substantial non-linearity in the past trends in life expectancy in recent decades, many of the approaches currently used rely on a shorter past period (Stoeldraijer et al. 2013a), or rely on the trends for women that were more linear (e.g. Bergeron-Boucher et al. 2018). Moreover, it is in cases in which the past trends were non-linear that the projection outcomes rely most heavily on the historical time period chosen (Stoeldraijer et al. 2013a). Because our approach relies on the extrapolation of non-lifestyle-attributable mortality, which exhibits more linear past trends than all-cause mortality (especially from a historical point of view)(see Figure 2 and Figure 2 —figure supplement 1)(=previous Figure S2), the effect of the choice of the historical time period is smaller than it is in extrapolations of all-cause mortality. However, including only the trends from 1990 onwards has likely resulted in an underestimation of the effects of the separate extrapolation of non-lifestyle-attributable mortality and of lifestyle-attributable mortality, as for men in particular, the differences between the trends in e0 for all-cause and for non-lifestyle-related mortality would have been much larger if a longer past trend had been included. This is primarily because of the large increases in smoking-attributable mortality among men in many Western European countries, which resulted in stagnating increases in life expectancy in the 1960s and 1970s (see Figure 2); and because of the large increases in alcohol-attributable mortality among Eastern European men since 1975 (e.g. Trias-Llimós et al. 2017), which contributed as well to the past stagnating trends in life expectancy in these regions (Meslé et al. 2002).

We have now added a justification for our decision to use a relatively long projection horizon given the comparatively short historical period (see lines 170-173; appendix 1 – supplementary data and methods – page 2 paragraph 1). In addition, in the discussion, we explained that our approach – because it is focused on non-lifestyle-attributable mortality – is less dependent on the choice of the historical period (lines 360 – 363 and lines 443- 446).

The forecasting of lifestyle-attributable mortality indeed involves more subjectivity than the extrapolation of non-lifestyle-attributable mortality, as we acknowledge in our paper (see lines 419-421). To avoid generating unrealistic outcomes for the (long-term) future (e.g., continuing increases in smoking-attributable mortality for women, in obesity-attributable mortality, and in alcohol-attributable mortality in some Eastern European countries), we had to adopt approaches that did not simply extrapolate past trends, but that also took into account knowledge about the likely future progression of smoking-, alcohol-, and obesity-attributable mortality. In doing so, however, we still made as much use as possible of the past trends (for example, by making use of past trends for men in the case of smoking and of past trends in other European countries in the case of alcohol; and by adopting an extrapolation approach, but then applying it to the speed of change in the logit of prevalence for obesity to ensure a wave-shaped pattern, and by applying a quadratic curve to the past trends for populations that were still experiencing increases in smoking and obesity), and implemented our approach in frequently used models (e.g., the Lee-Carter model and age-period-cohort models). See Appendix 1 – Box 1 for a more extended summary of the different approaches used to project obesity-, smoking-, and alcohol-attributable mortality; and see Janssen, van der Broek et al., 2020; Janssen, El Gewily, Bardoutsos, 2020; and Janssen, El Gewily et al., 2020 for the actual projections, including a critical appraisal of each of them.

We have now extended the discussion of the projection of lifestyle-attributable mortality (see lines 413 – 422):

“To avoid generating unrealistic outcomes for the (long-term) future (e.g. continuing increases in smoking-attributable mortality for women, in obesity-attributable mortality, and in alcohol-attributable mortality in the few Eastern European countries where it is still increasing), the projections were based on advanced approaches that did not simply extrapolate past trends, but also took into account knowledge about the future progression of smoking-, alcohol-, and obesity-attributable mortality. Nonetheless, the projection of lifestyle-attributable mortality involves more subjectivity than the extrapolation of non-lifestyle-attributable mortality (see Janssen, van der Broek et al., 2020; Janssen et al., 2020; Janssen, El Gewily and Bardoutsos, 2020 for a critical evaluation)”.

Finally, the principal added value of the manuscript – the new approach to forecast mortality – is currently not available to the general public: "Because of the large size of the many different underlying simulation matrices, we are restricted in sharing these data. … The R codes can be requested from the author." One simple way to address this problem is to share a much smaller set of simulations (say 100 instead of 50,000) and warn the users to increase the number of simulations when running the codes on their machines. This would directly provide the research community with the routines to employ the proposed methodology.

We carefully considered what data we could share in order to make the new approach to forecasting mortality available to the general public. Although much of the information is already provided on Open Science Framework (https://osf.io/ghu45/) in the Excel file with the final data used as input for the tables and figures, we will now also provide information on the underlying observed age-specific mortality rates (all-cause mortality, non-lifestyle-attributable mortality, lifestyle-attributable mortality), as well as the fitted and projected age-specific mortality rates (by means of medians and 90% and 95% projection intervals). In addition, we will make the different R codes used for the different steps of the data analyses public.

In line with this change, we rewrote our data sharing statement: “Some of our original data regarding lifestyle-attributable mortality were based on previous publications, which, in turn, used data that are openly available. The all-cause mortality data and the exposure data can be obtained through the Human Mortality Database. We have provided source data files for all our tables and figures. These comprise the numerical data that are represented in the different figures, and the output on which the different tables are based. In addition, one excel file with all the final numerical / output data that were used as input for the tables and figures will be made available at the Open Science Framework: https://osf.io/ghu45/. In addition, we will upload there the underlying observed age-specific mortality rates (all-cause mortality, non-lifestyle-attributable mortality, lifestyle-attributable mortality) as well as the adjusted and projected age-specific mortality rates (medians and 90% and 95% projection intervals). The different R codes used for the different steps of the data analyses will be shared – as well – through the Open Science Framework link.

ReferencesBergeron-Boucher, M.-P., V. Canudas-Romo, J. Oeppen, and J. W. Vaupel (2017). Coherent forecasts of mortality with compositional data analysis. Demographic Research 37, 527-566.Bohk-Ewald, C., M. Ebeling, and R. Rau (2017). Lifespan Disparity as an Additional Indicator for Evaluating Mortality Forecasts. Demography 54 (4), 1559-1577.Lee, R. D. and L. R. Carter (1992). Modeling and forecasting US mortality. Journal of the American Statistical Association 87 (419), 659-671.Shang, H. L., H. Booth, and R. Hyndman (2011). Point and interval forecasts of mortality rates and life expectancy: A comparison of ten principal component methods. Demographic Research 25, 173-214.van Raalte, A. A., I. Sasson, and P. Martikainen (2018). The case for monitoring life-span inequality. Science 362 (6418), 1002-1004.Reviewer #2 (Recommendations for the authors):The authors have conducted an ambitious projection exercise, drawing together a number of methods and approaches to seek to understand the impacts of future changes in lifestyle- and non-lifestyle attributable mortality on prospects for life expectancy increase in a set of European countries. This focus on the underlying dynamics of mortality change--that is, disaggregating the long-term changes in non-lifestyle attributable mortality patterns from the much more variable lifestyle-attributable mortality patterns--represents a real step forward in life expectancy projection modeling. These methods could have considerable benefits for projections in other contexts.My main issue with the paper as written concerns the assumptions built into the Li-Lee projection of non-lifestyle attributable mortality, where I would encourage the authors to test the sensitivity of their projections to alternative choices of reference countries (looking beyond Italy, Spain, and France).

Yes, the choice of the reference or forerunner populations is indeed an important aspect of multi-population or coherent mortality forecasting (Stoeldraijer 2019; Booth 2020). However, given that the trend in non-lifestyle-attributable mortality is more equal between countries and sexes than the trend in all-cause mortality (see Table 1), this choice will likely have a larger impact when coherently forecasting all-cause mortality than when coherently forecasting non-lifestyle-attributable mortality (see also Janssen and Kunst 2007 and Janssen et al. 2013). We selected women in France, Spain, and Italy as the forerunner populations in terms of life expectancy in Europe because they exhibit both very high recent life expectancy values and very favourable long-lasting past trends in e0 (measured by large increases in e0 over the 1970-2011 period among non-Eastern European countries) (Stoeldraijer 2019). Our decision to use forerunner populations as the reference populations is in line with the seminal work by James Vaupel (e.g., Oeppen and Vaupel 2002), and our decision to use female populations is in line with the recent work by Bergeron-Boucher et al. 2018. In addition, using three populations instead of one is regarded more robust.

An alternative approach would be to use the average mortality experience across many countries (Eurostat 2020; Stoeldraijer 2019), or – as indicated by reviewer 2 in the extended (non-public) explanation of this remark – to use the mortality experience of a set of low-mortality countries that have faced less severe life-course mortality pressures--e.g. Sweden, Norway, and Switzerland.

Author response table 1 reports on the outcomes of a sensitivity analysis in which we employed these two alternative selections of the reference population in coherent mortality projection. Our sensitivity analysis revealed that the use of the average mortality experience across all the included countries resulted – in line with our expectations – in lower projected e0 values in 2065, for both non-lifestyle attributable mortality and all-cause mortality. Furthermore our sensitivity analysis revealed that the use of women in Sweden, Norway and Switzerland as the reference population led to lower projected e0 values in 2065, but particularly so for all-cause mortality and much less so for non-lifestyle-attributable mortality. This latter outcome can be largely explained from the difference in the average (unweighted) gains in e0 over the 1990-2014 period for women in France, Italy, and Spain (4.8 for all-cause mortality; 5.0 for non-lifestyle-attributable mortality) compared to women in Sweden, Norway, and Switzerland (4.1 for all-cause mortality; 4.8 for non-lifestyle-attributable mortality).

The results of our sensitivity analysis, confirm that the use of forerunner populations instead of an average across many populations results in lower projected e0 values in 2065, and that the choice of the reference population will likely have a larger impact when coherently forecasting all-cause mortality than when coherently forecasting non-lifestyle-attributable mortality.

We have added a short discussion on the choice of the reference populations in the “Appraisal of our projection methodology” section of our revised manuscript (lines 384 – 396):

“However, as well as the potential effects on the outcomes of the exact coherent methodology that is applied (Stoeldraijer, 2019), it is important to take into account the choice of the reference population (Stoeldraijer 2019; Booth 2020). Using forerunner populations as the reference population, as advocated by among others Oeppen and Vaupel 2002 and Bergeron-Boucher et al., 2018, instead of an average across many populations (Li and Lee 2005; Eurostat 2020) will generally result in higher future life expectancy values (see the next section as well). However, given that the past trend in e0 for non-lifestyle-attributable mortality is more equal between countries and sexes than the past trend in e0 for all-cause mortality (see Table 1), which reference populations are chosen will likely have a larger impact when coherently forecasting all-cause mortality than when coherently forecasting non-lifestyle-attributable mortality (see also Janssen and Kunst 2007 and Janssen et al., 2013)”.

References

Edwards, R. D. and Tuljapurkar, S. (2005). Inequality in life spans and a new perspective on mortality convergence across industrialized countries. *Population and Development Review*, 31(4):645–674

Garcia, J., and Aburto, J.M. (2019). The impact of violence on Venezuelan life expectancy and lifespan inequality. *International Journal of Epidemiology*, 48(5), 1593-1601.

Meslé, F. and J. Vallin (2002). Mortality in Europe: The divergence between East and West. *Population*, 57: 157-197.

Technical Panel on Assumptions and Methods (2003). Report to the Social Security Advisory Board. Washington, D.C.

Trias-Llimós, S., Bijlsma, M. And Janssen, F. (2017). The role of birth cohorts in long-term trends in liver cirrhosis mortality across eight European countries. *Addiction* ,112(2), 250-258.